# ZNF827 is a single-stranded DNA binding protein that regulates the ATR-CHK1 DNA damage response pathway

Sile F. Yang[1], Christopher B. Nelson[1], Jadon K. Wells [1], Madushan Fernando[1], Robert Lu[1], Joshua A. M. Allen[1], Lisa Malloy[1], Noa Lamm[2], Vincent J. Murphy [3], Joel P. Mackay [4], Andrew J. Deans [3,5], Anthony J. Cesare [6], Alexander P. Sobinoff [1] & Hilda A. Pickett [1] ✉

The ATR-CHK1 DNA damage response pathway becomes activated by the exposure of RPA-coated single-stranded DNA (ssDNA) that forms as an intermediate during DNA damage and repair, and as a part of the replication stress response. Here, we identify ZNF827 as a component of the ATR-CHK1 kinase pathway. We demonstrate that ZNF827 is a ssDNA binding protein that associates with RPA through concurrent binding to ssDNA intermediates. These interactions are dependent on two clusters of C2H2 zinc finger motifs within ZNF827. We find that ZNF827 accumulates at stalled forks and DNA damage sites, where it activates ATR and promotes the engagement of homologous recombination-mediated DNA repair. Additionally, we demonstrate that ZNF827 depletion inhibits replication initiation and sensitizes cancer cells to the topoisomerase inhibitor topotecan, revealing ZNF827 as a therapeutic target within the DNA damage response pathway.

Genome stability is constantly being threatened by assaults from endogenous and exogenous genotoxic agents, as well as intrinsic replication obstacles[1–3]. These insults can result in replication stress, which is defined as alterations in the rate of replication fork progression. DNA lesions that cause replication stress carry catastrophic consequences, as failure to resolve such lesions results in replication fork collapse and DNA double-strand breaks (DSBs)[4,5]. Consequently, robust replication stress response mechanisms exist to preserve genome stability and ensure cell survival. Ataxia telangiectasia mutated and Rad3-related (ATR) is a phosphoinositide 3-kinase-related protein kinase that functions as the master regulator of the replication stress response[3,6,7]. Specifically, the ATR-mediated DNA damage response (DDR) signalling pathway is essential for stabilizing stalled forks, preventing fork collapse, promoting fork restart through homologous recombination (HR)-directed repair

mechanisms, and repairing one-ended DNA breaks arising from collapsed replication forks[3,6].

The ability of ATR to sense genomic DNA damage is dependent on replication protein A (RPA)[8]. RPA is a heterotrimeric protein complex, comprising RPA70, RPA32 and RPA14 subunits, that binds and stabilizes exposed ssDNA. This includes ssDNA intermediates formed during normal replication, as well as at stalled replication forks and at resected DSBs[9,10]. In addition to RPA, other eukaryotic ssDNA binding proteins (SSBs) include RAD51, POT1 and hSSB1, RADX and ZPET[11–13]. SSBs play essential roles in genomic metabolic processes by stabilizing ssDNA intermediates, recruiting downstream targets, and facilitating enzymatic activities at normal and stressed replication forks, sites of DNA damage, and at telomeres[10,14–16].

ATR localizes to ssDNA-RPA complexes through its obligate binding partner, ATR-interacting protein (ATRIP)[8,17]. ATR-ATRIP

[1]Telomere Length Regulation Unit, Children's Medical Research Institute, Faculty of Medicine and Health, University of Sydney, Westmead, NSW 2145, Australia. [2]Nuclear Dynamics Group, Children's Medical Research Institute, Faculty of Medicine and Health, University of Sydney, Westmead, NSW 2145, Australia. [3]Genome Stability Unit, St Vincent's Institute, Fitzroy, VIC 3065, Australia. [4]School of Life and Environmental Sciences, University of Sydney, Sydney, NSW 2006, Australia. [5]Department of Medicine (St Vincent's), University of Melbourne, Fitzroy, VIC 3065, Australia. [6]Genome Integrity Unit, Children's Medical Research Institute, Faculty of Medicine and Health, University of Sydney, Westmead, NSW 2145, Australia. ✉e-mail: hpickett@cmri.org.au

activation requires binding of an additional activator protein. Two ATR activators have been identified in vertebrates, with defined ATR activating domains (AADs)[18]. Topoisomerase II binding protein 1 (TOPBP1) is recruited to DNA lesions with 5′-ended ssDNA-double-stranded DNA (dsDNA) junctions through its interaction with the RAD9-RAD1-HUS1 (9-1-1) checkpoint clamp and the MRE11-RAD50-NBS1 (MRN) complex, and stimulates ATR kinase activities by direct binding to the ATR-ATRIP complex[19,20]. More recently, Ewing tumour-associated antigen 1 (ETAA1) was identified as a second ATR activator. ETAA1 is recruited to RPA-coated ssDNA through a direct interaction with RPA[21–23]. This regulates ATR signalling through the S/G2 transition to ensure completion of replication prior to transactivation of the mitotic gene network[24].

Once activated, ATR phosphorylates a host of downstream targets, most notably the CHK1 effector kinase, which in turn phosphorylates CDC25A and inhibits cyclin-dependent kinase (CDK) activity, thereby triggering activation of the intra-S phase and G2/M phase checkpoints[25–27]. This ultimately regulates stalled fork repair and, if necessary, firing of dormant replication origins. ATR activation also promotes HR-mediated DNA repair. HR is a high-fidelity repair mechanism that is reliant on DNA sequence homology to provide the repair template, thereby restricting its role to the S and G2 phases of the cell cycle. HR is initiated by ssDNA, and involves strand invasion of the single-stranded sequence into complementary duplex DNA, template-directed repair, and substrate resolution[1]. Our understanding of the ATR activation model and the intricacies of the stress response continue to expand and be refined as novel ATR regulators and modulators are identified[3,7].

ZNF827 is a member of the C2H2 zinc finger (ZnF) protein superfamily. C2H2 zinc finger motifs typically cluster, with the organisation of these clusters conferring DNA recognition and binding[28]. ZNF827 has consistently been identified at telomeres specifically engaged in rampant HR-mediated repair events as part of the Alternative Lengthening of Telomeres (ALT) pathway[29–32]. In this capacity, ZNF827 recruits the nucleosome remodelling and deacetylase (NuRD) complex through a conserved N-terminal RRK motif to promote telomere recombination, whilst simultaneously maintaining the telomere substructure[33,34]. Here, we define an additional function of ZNF827 in the genomic DDR. We find that ZNF827 is a ssDNA binding protein that binds to ssDNA in parallel with RPA. ZNF827 depletion inhibits replication initiation and fork progression, and results in cell cycle arrest at the G1/S transition. ZNF827 accumulates at stalled replication forks and sites of DNA damage, interacts with TOPBP1, ETAA1, ATRIP and ATR, and functions in the activation of the ATR-CHK1 DDR pathway to regulate HR-mediated DNA repair.

## Results

### ZNF827 is a ssDNA binding protein that associates with RPA

ZNF827 contains nine C2H2 ZnF motifs spatially arranged in two clusters. The first (ZnF1-3) is located centrally, and the second (ZnF4-9) is located in the C-terminal region (Fig. 1A). ZNF827 also contains an N-terminal RRK domain that shows conserved binding to the NuRD complex[33,34] (Fig. 1A). To investigate the DNA binding capabilities of ZNF827, we purified ZNF827 recombinant protein. Protein expression was confirmed by HaloTag® TMR ligand detection, and protein purification was validated by Coomassie staining, western blot analysis and mass spectrometry (MS) (Supplementary Fig. 1A, B; Supplementary Table 1). Electrophoretic mobility shift assays (EMSAs) demonstrated concentration-dependent binding of purified ZNF827 to a ssDNA pentaprobe[35], indicated by the gel shift pattern (Fig. 1B). No binding was observed between ZNF827 and double-stranded (ds) DNA (Fig. 1B). Bound complexes were resolved as two distinct bands, indicative of ZNF827 and ssDNA forming complexes with different stoichiometries, depending on the availability of ZNF827. A super-shift was observed in the presence of ZNF827 antibody, supporting the

specificity of ZNF827 binding to ssDNA (Supplementary Fig. 1C). Binding of ZNF827 to ssDNA was further demonstrated by immunofluorescence (IF) using a direct ZNF827 antibody, which identified distinct colocalizations between ZNF827 and ssDNA, visualized by native 5-bromo-2′-deoxycytidine (BrdU) foci (Fig. 1C). Finally, ZNF827 binding to ssDNA was demonstrated by incubation of purified protein with either ssDNA or dsDNA, followed by immunoprecipitation (IP) using a direct ZNF827 antibody (Supplementary Fig. 1D).

To elucidate the mechanism of DNA binding, we generated ZNF827 mutant proteins in which either ZnF cluster was deleted (ZnFΔ1-3, ZnFΔ4-9), and a severely truncated ZNF827 mutant protein, which includes the removal of both ZnF domains (Truncated) (Fig. 1A and Supplementary Fig. 1E). EMSAs using purified wild-type and mutant proteins revealed that ssDNA binding was obliterated following incubation with the ZnFΔ1-3 and Truncated proteins, while the ZnFΔ4-9 mutant protein retained only residual ssDNA binding capabilities (Fig. 1D). None of the mutant proteins exhibited binding activity to dsDNA (Fig. 1D). These data demonstrate that the ZNF827 ZnF domains, and primarily the central cluster of C2H2 motifs, are required for ssDNA binding.

We next investigated the association between ZNF827 and the major ssDNA binding protein complex RPA. Co-immunoprecipitation (co-IP) experiments demonstrated that RPA32 and RPA70 associated with ZNF827 in U-2 OS and HT1080 cells (Fig. 1E and Supplementary Fig. 1F). However, the interaction was destroyed following treatment with the nuclease benzonase, thereby precluding a direct interaction between the two proteins (Fig. 1E). NuRD components HDAC1 and RBBP4, which are known to bind directly to ZNF827 through its N-terminal RRK domain[33,34], were included as positive controls and retained binding following treatment with benzonase (Fig. 1E). Consistent with the association between ZNF827 and RPA being mediated by ssDNA, co-IP was predominantly dependent on the central cluster of C2H2 motifs, and to a lesser extent the C-terminal cluster (Fig. 1F).

We then used IF to visualize the cellular localization of ZNF827, and the association between ZNF827 and RPA. Comparable exogenous expression of Myc-tagged ZNF827 constructs in U-2 OS CRISPR-Cas9 ZNF827 knockout cells (Supplementary Fig. 1E) demonstrated prominent punctate nuclear colocalizations of wild-type ZNF827 with RPA32 (Fig. 1G). The ZNF827 ZnFΔ1-3 mutant showed disperse cytoplasmic as well as nuclear staining and a limited number of colocalizations with RPA, indicative of this mutant being partially defective in nuclear localisation (Fig. 1G). ZNF827 ZnFΔ4-9 displayed diffuse nuclear staining and did not form foci with RPA, while the ZNF827 Truncated mutant showed a similar staining pattern to ZNF827 ZnFΔ1-3 with no colocalization with RPA. The ΔRRK mutant retained nuclear localization and association with RPA, indicative of the interaction between ZNF827 and NuRD being dispensable for cellular localization and ssDNA binding (Fig. 1G).

To further elucidate the dynamics of ZNF827 and RPA binding to ssDNA we performed comparative and competitive EMSAs using purified ZNF827 and RPA protein complex at concentrations such that ssDNA binding was saturated (with no additional free probe available). As expected, RPA bound robustly to ssDNA, visualized by a clear single gel shift product (Fig. 1H). ZNF827 consistently bound to ssDNA independently of RPA. Simultaneous addition of ZNF827 and RPA resulted in a reduction of both ZNF827-ssDNA and RPA-ssDNA complexes, accompanied by the appearance of an additional band that resolved between the RPA and ZNF827 gel shift bands (Fig. 1H). This additional band was observed only when both ZNF827 and RPA were present, indicative of an intermediary complex formed when both proteins bind to the same ssDNA probe. Addition of RPA to the ZNF827-ssDNA reaction resulted in substantial disruption of ZNF827-ssDNA complexes, accompanied by formation of intermediary complexes as well as RPA-ssDNA (Fig. 1H), indicative of some displacement of ZNF827 by RPA. In comparison, addition of ZNF827 to the RPA-ssDNA reaction favoured retention of RPA-ssDNA complexes, with

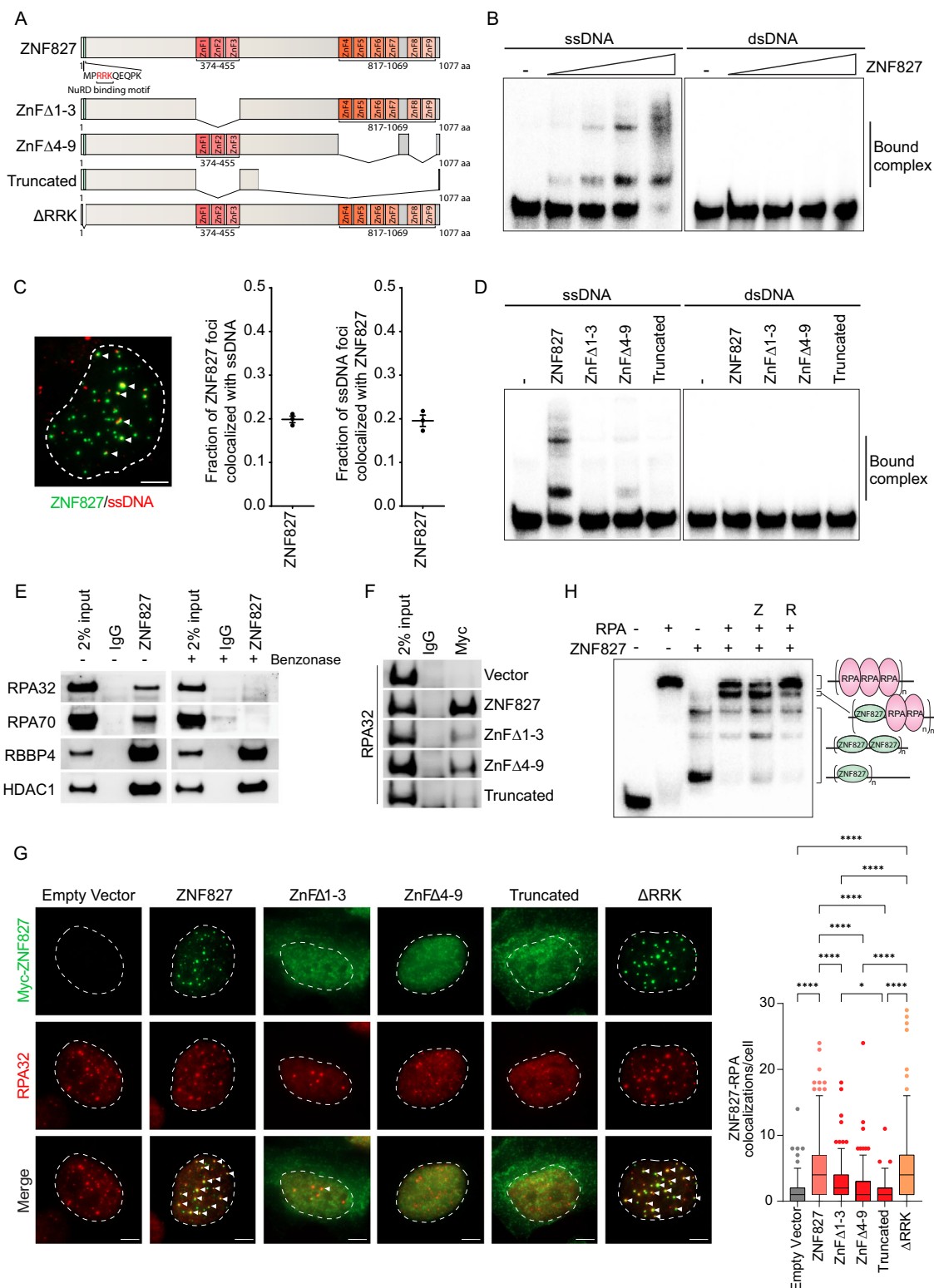

only a partial shift to intermediary complexes and ZNF827-ssDNA (Fig. 1H). Overall, these data demonstrate that ZNF827 binds to ssDNA concurrently with RPA and that binding is mediated by the C2H2 ZnF domains within ZNF827.

## ZNF827 accumulates at sites of replication stress-associated DNA damage

RPA localizes to ssDNA that forms as an intermediate during normal replication, and accumulates at stalled forks in response to

replication stress[36–39], while proliferating cell nuclear antigen (PCNA) functions as a sliding clamp to anchor the replication machinery to the DNA[40]. We next investigated whether ZNF827 localizes with replication forks during their progression. Live cell imaging demonstrated persistent associations between fluorescence-labelled HaloTag-ZNF827 and both GFP RPA32 (Supplemental Movie 1) and PCNA chromobodies (Supplemental Movie 2) in U-2 OS cells. Co-IP of ZNF827 identified an interaction with both ATRIP, the regulatory partner of ATR, and 9-1-1 DNA damage sensor complex component

**Fig. 1 | ZNF827 binds to ssDNA and associates with RPA. A** Schematic of ZNF827 domains and functional mutants. **B** Electrophoretic mobility shift assay (EMSA) using γ-$^{32}$P radiolabelled single-stranded and double-stranded pentaprobes with increasing concentrations of purified ZNF827 protein ($n = 1$). Bound complex is indicated by the gel shift. **C** Representative image of ZNF827 (green) and native BrdU-stained ssDNA (red) colocalizations in U-2 OS cell nuclei following transient overexpression of ZNF827 (left panel). White arrows indicate colocalizations. Scale bar represents 5 μm. Manual quantitation of ZNF827 and ssDNA foci per nuclei and the fraction of colocalizing foci per nuclei (right panels). Data represent 50 nuclei from three biological replicates. **D** EMSA using γ-$^{32}$P radiolabelled single-stranded and double-stranded pentaprobes with purified ZNF827 wild-type and ZNF827 ZnF mutant proteins ($n = 1$). Bound complex is indicated by the gel shift. **E** Western blot analysis of RPA components, RPA32 and RPA70, and NuRD components, RBBP4 and HDAC1, following co-immunoprecipitation (co-IP) with a direct ZNF827 antibody in U-2 OS cells overexpressing ZNF827 ($n = 1$). **F** Western blot analysis of

RPA32 following co-IP with a Myc antibody in U-2 OS cells overexpressing Myc-tagged ZNF827 and ZNF827 ZnF mutants ($n = 1$). Empty vector included as a negative control. **G** Representative images of Myc-tagged ZNF827 and ZNF827 mutants (green) and RPA32 (red) immunofluorescence labelling in U-2 OS ZNF827 knockout cells 48 h after transient transfection with Myc-tagged ZNF827 constructs (left panel). White arrows indicate colocalizations. Scale bar represents 5 μm. Manual quantitation of ZNF827 foci colocalizing with RPA foci (right panel). Data represent 50 nuclei from three biological replicates and data from individual nuclei were plotted as Tukey box plots; ****$P < 0.0001$, *$P = 0.05$. Multiple comparisons were corrected for using the Bonferroni test. **H** EMSA using γ-$^{32}$P radiolabelled single-stranded pentaprobe with purified ZNF827 and RPA protein complex ($n = 1$). Z denotes addition of ZNF827 15 min before RPA. R denotes addition of RPA 15 min before ZNF827. Bound complex is indicated by the gel shift, and the associated schematic indicates possible protein binding configurations. Source data are provided as a Source Data file.

Rad9 (Supplementary Fig. 2A), indicative of ZNF827 enrichment at stalled forks.

To directly measure the effects of ZNF827 depletion on DNA replication fork progression, we used single-molecule analysis of replicating DNA (SMARD) following siRNA-mediated ZNF827 knockdown (Supplementary Fig. 2B). Cells were labelled with IdU for 30 min followed by CldU for 60 min (Fig. 2A). SMARD revealed a significant reduction in both replication rate and the frequency of second origins, defined as new origin of replication activation events, following ZNF827 depletion (Fig. 2B), indicative of a shut-down of replication programs within the cell. This was accompanied by a moderate increase in the incidence of stalled replication forks (Fig. 2B). These data demonstrate that ZNF827 promotes replication initiation and fork progression.

To determine the role of ZNF827 in the DNA damage response, cells were treated with topotecan, a camptothecin analogue that traps topoisomerase I on DNA, thereby causing replication stress and the consequent induction of ssDNA intermediates and DSBs. As expected, topotecan treatment resulted in an induction of nuclear γH2AX foci, a marker of DNA damage (Fig. 2C, Supplementary Fig. 2D, E). Similarly, topotecan treatment caused an increase in nuclear ZNF827 staining. ZNF827 colocalised with γH2AX foci in untreated cells, and this interaction was exacerbated upon DNA damage induction following topotecan treatment in U-2 OS, HT1080 and HT10806TG cells (Fig. 2C, Supplementary Fig. 2D, E). Depletion of RPA32 (Supplementary Fig. 2F) also resulted in an induction of nuclear γH2AX foci, as well as increased recruitment of ZNF827 to DNA damage sites, but no overall change in cellular ZNF827 levels was observed (Fig. 2D).

Laser micro-irradiation identified colocalization of ZNF827 with both γH2AX and RPA32 at laser stripes in HT1080 cells (Supplementary Fig. 2G), further reinforcing the localization of ZNF827 to sites of DNA damage. Together, these data show that ZNF827 is recruited to sites of DNA damage concomitantly, but independently of RPA, and suggest that ZNF827 recruitment to ssDNA may compensate for RPA loss or exhaustion.

To further verify the recruitment of ZNF827 to sites of replication stress-induced DNA damage, we utilized FANCM depletion in U-2 OS cells as a means to generate excessive levels of telomere-specific replication stress and telomeric ssDNA[41,42]. FANCM depletion resulted in an increase in both RPA32 and ZNF827 colocalizations with telomeric DNA, and an increase in colocalizations between ZNF827, RPA and telomeres (Fig. 2E), directly demonstrating the recruitment of ZNF827 to sites of replication stress. Overall, these data indicate that ZNF827 accumulates at sites of replication-associated DNA damage.

## ZNF827 activates the ATR-CHK1 kinase pathway in response to replication stress

The ATR-CHK1 kinase pathway responds to stalled replication forks to coordinate cell cycle arrest, fork repair and HR-mediated DNA

repair. We next investigated whether ZNF827 was involved in ATR-mediated DNA damage signalling. In unperturbed U-2 OS cells, ZNF827 depletion caused a reduction in both CHK1 levels and phosphorylation of CHK1 S345 (Fig. 3A). In cells treated with topotecan and allowed to recover for 1 h, we detected CHK1 phosphorylation at S345 and RPA phosphorylation at S33, indicative of robust ATR activation (Fig. 3A). Strikingly, phosphorylation of both CHK1 S345 and RPA32 S33 was almost entirely suppressed following ZNF827 depletion, despite topotecan-induced replication-associated DNA damage persisting, as demonstrated by elevated levels of γH2AX in both scrambled and ZNF827 depleted topotecan-treated cells (Fig. 3A).

Suppression of ATR-CHK1 kinase pathway activation was confirmed using three additional siRNAs targeting ZNF827 (Supplementary Fig. 2B, Supplementary Fig. 3A). Complementation experiments using an siRNA targeting the 3′ UTR of ZNF827 demonstrated partial rescue of the ATR pathway defect in U-2 OS cells stably overexpressing wild-type ZNF827, but not with the ZNF827 ZnFΔ1-3 mutant (Fig. 3B, Supplementary Fig. 3B, C). This demonstrates a specific role for ZNF827 in ATR pathway activation, and is consistent with ZNF827 playing a role in ATR-CHK1 signalling through its ssDNA binding capability. Notably, during the course of these experiments, we observed that stably overexpressed wild-type ZNF827 leads to detection of a smear by western blot analysis. ZNF827 depletion also resulted in a significant decrease in mean pRPA foci/RPA foci in S-phase cells following treatment with topotecan (Fig. 3C), supporting impaired ATR-CHK1 pathway activation. This observation precludes the ATR pathway activation defect caused by ZNF827 depletion being entirely attributable to cell cycle perturbation.

By extending the recovery time following topotecan treatment to 24 h, we observed suppression of ATR-CHK1 kinase pathway activation upon ZNF827 depletion in four cell lines (U-2 OS, HT1080, HT10806TG and IIICF/c) (Fig. 3A, Supplementary Fig. 3D–F). Some variation in the temporal response to ATR pathway activation in the context of ZNF827 depletion was detected across the different cell lines; however, robust suppression of ATR signalling was observed within 24 h of topotecan treatment. The inhibitory effects on phosphorylation of CHK1 S345 and RPA32 S33 were also observed at an earlier timepoint (24 h) post-transfection with ZNF827 siRNA when effective knockdown was verified (Supplementary Fig. 2B, Supplementary Fig. 3G). This is indicative of the suppression of ATR pathway activation being attributable to ZNF827 depletion, rather than being an indirect effect of the downstream cellular consequences of ZNF827 depletion.

To characterise the mechanism of ATR activation, we determined whether overexpression of the ATR activators TOPBP1 or ETAA1 was able to overcome the suppressive effects of ZNF827 depletion on ATR activation. TOPBP1 overexpression partially restored ATR pathway activation, as indicated by increased levels of pCHK1 S345 and pRPA32 S33 in ZNF827 depleted cells compared to empty vector control

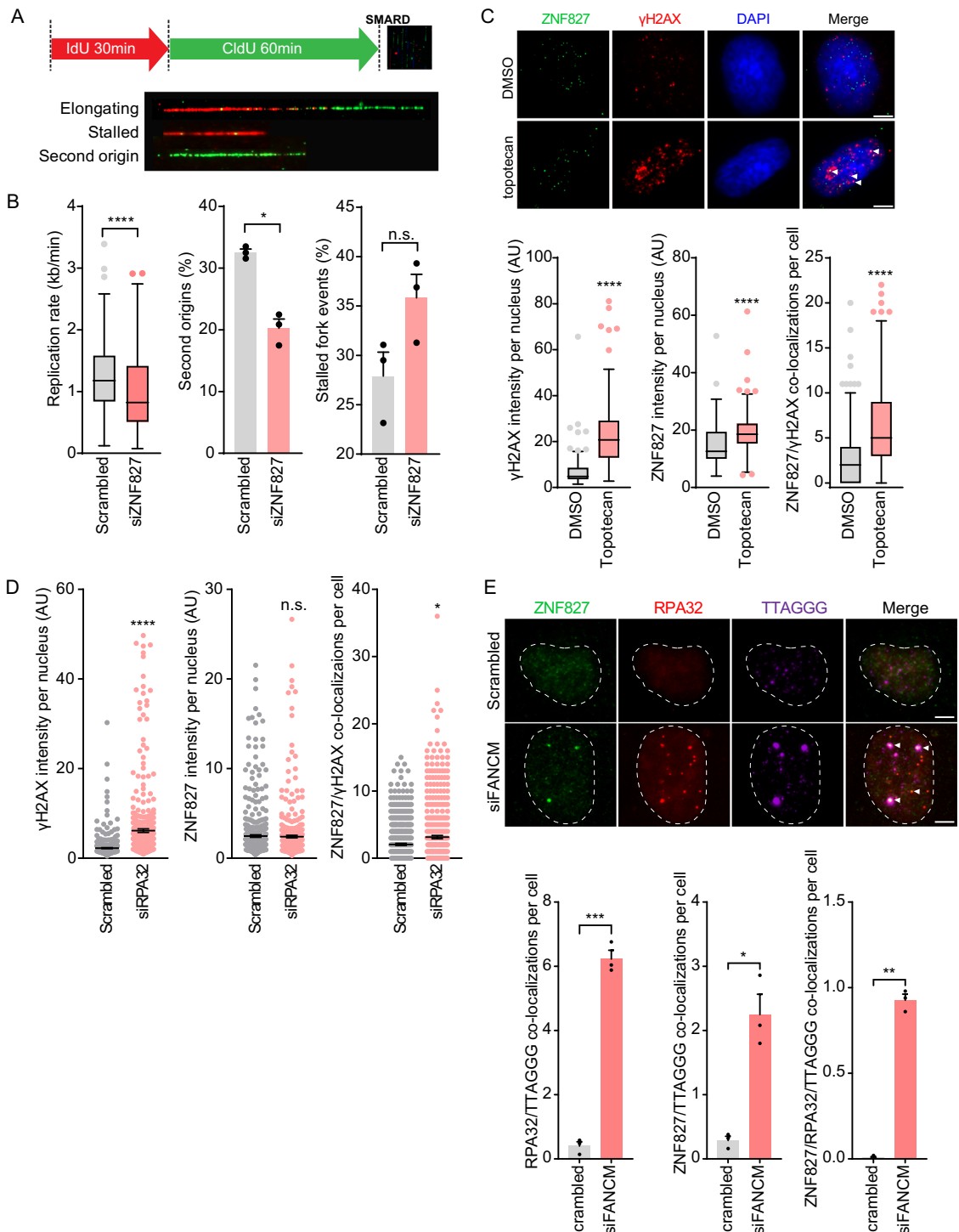

(Fig. 3D). In contrast, ETAA1 overexpression was unable to rescue any ATR activation in ZNF827 depleted cells (Fig. 3E). This demonstrates that the suppressive effects of ZNF827 depletion on the ATR pathway can be rescued by overexpression of TOPBP1.

We then investigated the potential for a direct interaction between ZNF827 and ATR. Co-IP experiments demonstrated that ATR, TOPBP1, ETAA1 and ATRIP, all co-immunoprecipitate with ZNF827 (Fig. 3F, G). Binding of TOPBP1 and ATR to ZnFΔ1-3 was also observed (Fig. 3F), indicative of the interaction being independent of both ZNF827 binding to ssDNA and the association between ZNF827 and RPA. The ZNF827 ΔRRK mutant also retained binding to TOPBP1 and ATR, but as expected, did not bind to NuRD components HDAC1 and

MTA1 (Fig. 3F). Intriguingly, the interaction between ZNF827 and TOPBP1 appeared to be enhanced in the absence of the central cluster of ZnF domains, indicative of allosteric competition between TOPBP1 and ssDNA for binding to ZNF827. Overall, these data demonstrate that ZNF827 is a TOPBP1, ETAA1, ATR and ATRIP interactor that plays a direct role in activation of the ATR-CHK1 kinase pathway.

## ZNF827 depletion impairs HR-mediated repair and sensitises cancer cells to topotecan

The ATR-CHK1 pathway regulates HR-mediated repair of stalled and collapsed replication forks. To identify the effects of ZNF827 depletion on HR, we quantitated sister-chromatid exchange (SCE) events and

**Fig. 2 | ZNF827 accumulates at sites of replication-associated DNA damage.**
**A** DNA fibre assay and replication events. **B** Replication rate in U-2 OS cells following
ZNF827 depletion (left panel). Data represent $n = 191$ fibres in scrambled and $n = 197$
fibres in siZNF827 cells across three independent experiments plotted as Tukey box
plots; *$P = 0.0075$, ****$P < 0.0001$, n.s. not significant ($P = 0.08$) by two-tailed
unpaired Mann–Whitney or Welch's $t$-test. Percentage second origins (middle
panel) and percentage stalled fork events (right panel) in U-2 OS cells following
ZNF827 depletion. Data presented as mean + SEM from three independent
experiments with at least 200 fibres from each experiment. n.s. non-significant;
*$P < 0.05$ by two-tailed $t$-test. **C** Representative images of γH2AX (red) and endo-
genous ZNF827 (green) foci in U-2 OS cells (top panel). White arrows indicate
colocalizations. Scale bar represents 5 μm. Quantitation of γH2AX and ZNF827
integrated intensity per nucleus, and γH2AX and ZNF827 colocalizations per
nucleus (bottom panel) from $n = 508$ nuclei in DMSO and $n = 399$ nuclei in
topotecan-treated cells using CellProfiler. At least 50% overlap criterion was applied
for colocalization count. Data are representative of two independent experiments
and presented as Tukey box plots; ****$P < 0.0001$ by two-tailed unpaired
Mann–Whitney tests. **D** Quantitation of γH2AX and ZNF827 integrated intensity per
nucleus, and ZNF827 and γH2AX colocalizations per nucleus in U-2 OS cells 72 h
post RPA32 knockdown. Out of three experiments, n = 493 nuclei in scrambled and
$n = 431$ nuclei in siRPA32 treated cells were scored. Data presented as mean ± SEM;
****$P < 0.0001$, *$P = 0.01$, n.s. not significant ($P = 0.23$) by two-tailed unpaired
Mann–Whitney tests. **E** Representative images of endogenous ZNF827 (green),
RPA32 (red) and telomeric DNA (purple) in U-2 OS cells 72 h post FANCM knock-
down (top panel). White arrows indicate colocalizations. Scale bar represents 5 μm.
Quantitation of RPA32 and ZNF827 colocalizations with telomeres, and RPA32,
ZNF827 and telomere colocalizations from at least 150 nuclei per replicate (bottom
panel). At least 50% overlap criterion was applied for colocalization count. Data are
representative of three independent experiments and presented as mean + SEM;
*$P = 0.0251$, **$P < 0.0011$, ***$P = 0.0004$ by two-tailed unpaired Welch's $t$-test. Source
data are provided as a Source Data file.

performed direct HR reporter assays. U-2 OS ZNF827 CRISPR-knockout
cells were used specifically because they displayed continuous cell
proliferation and a comparable proliferation rate to wild-type cells,
that we attribute to an adaptive response in these cells that enables
their survival. No change in the number of SCE events was observed
when compared to parental U-2 OS cells (Fig. 4A, B). Topotecan
treatment induced SCEs in the parental cells, but this effect was sig-
nificantly reduced in the ZNF827 knockout cells (Fig. 4A, B). Similarly,
unperturbed HT1080 cells showed no change in SCE events following
ZNF827 depletion, while topotecan treatment resulted in a substantial
increase in SCEs that was significantly suppressed by ZNF827 deple-
tion (Fig. 4B).

HR reporter cell lines were created by stable transfection and
clonal selection of U-2 OS cells with DR-GFP[43]. Reporter assays, in
which HR repair is measured by the frequency of GFP gene restoration
following DSB induction by ISCE-I, revealed a significant decrease in
GFP-positive cells following ZNF827 depletion in U-2 OS cells over-
expressing the ISCE-I endonuclease (Fig. 4C). No GFP-positive cells
were observed in the absence of reporter activity by ISCE-I (Fig. 4C).
These data indicate that ZNF827 depletion inhibits the engagement of
HR-mediated DNA repair pathways.

To determine whether ZNF827 functions to recruit HR factors to
sites of genomic replicative damage, we performed IF to detect colo-
calizations between BRCA1 and pRPA32 in HT1080 cells in the context
of ZNF827 depletion (Fig. 4D). We identified a significant decrease in
BRCA1 foci per cell following ZNF827 depletion (Fig. 4E), and a cor-
responding significant decrease in colocalizations between BRCA1 and
pRPA32, that was independent of topotecan treatment (Fig. 4F). These
observations are consistent with NuRD-ZNF827-mediated recruitment
of HR factors to sites of replicative stress.

The reliance of cells on HR for the repair of DSBs, for instance
those caused by topoisomerase inhibitors, led us to investigate whe-
ther ZNF827 depletion conferred sensitivity to topotecan. Live cell
assays were performed in U-2 OS cells using the IncuCyte® system to
monitor real-time cell proliferation by confluency. Cells were treated
with topotecan or DMSO control at 24 h post-siRNA knockdown and
imaged every two hours. ZNF827 depletion significantly reduced cell
proliferation in U-2 OS cells, to a level comparable to that seen fol-
lowing topotecan treatment (Fig. 4G). ZNF827 depletion in conjunc-
tion with topotecan exacerbated the cell proliferation defect in a dose-
dependent manner, indicative of an additive effect (Fig. 4G). Similarly,
ZNF827 depletion significantly reduced cell proliferation in HT1080
cells, and combined ZNF827 depletion and topotecan treatment
obliterated cell growth over the duration of the experiment (Fig. 4H).
These data suggest that ZNF827 loss negatively impacts cell pro-
liferation and may sensitize cancer cells to DNA-damaging che-
motherapeutics such as topotecan.

## ZNF827 depletion results in p21 accumulation and G1/S cell cycle arrest

Finally, we aimed to characterize the cell proliferative defects con-
ferred by ZNF827 depletion. To determine the proliferative and repli-
cative effects of ZNF827 depletion we measured nuclear EdU
incorporation. A moderate decrease in the number of EdU-positive
cells was observed following ZNF827 depletion (Fig. 5A). More
noticeably, we observed a significant decrease in EdU intensity in the
S-phase cells following ZNF827 depletion (Fig. 5B), indicative of
ZNF827 depletion conferring an inhibitory effect on replication.

To directly assess changes to the cell cycle in response to ZNF827
depletion, we used live cell imaging. A cumulative increase in cell cycle
duration was identified in U-2 OS cells following ZNF827 depletion
(Fig. 5C). To further assess cell cycle dysregulation, we employed
fluorescence ubiquitination cell cycle indicator (FUCCI) HT1080 cells.
Live cell imaging of FUCCI cells identified a striking increase in G1
duration (red cells) following ZNF827 depletion (Fig. 5D), while no
impact on S/G2-M duration (green cells) was observed (Fig. 5E).
ZNF827 depletion also resulted in a significant proportion of cells
failing to progress correctly through mitosis (Fig. 5F–I). The majority of
these cells transitioned from G1 to early S phase, indicated by yellow
cells, and then reverted back to red (Fig. 5F, G), indicative of a failure
to commit to S phase, and reversion to either G1 or quiescence (G0).
A smaller proportion of cells similarly transitioned from G1 to early S,
reverted back to red and then underwent cell division, seemingly in G1
(Fig. 5F, H). Another proportion of cells transitioned from G2 to G1
without mitotic entry (Fig. 5F, I), thereby bypassing mitosis[44]. These
observations are consistent with ZNF827 depletion impeding normal
DNA replication, resulting in failure of S phase commitment and cell
cycle inhibition at G1/early S phase.

DNA damage is a major inhibitor of S phase commitment, and
expression of the cyclin-dependent kinase (CDK) inhibitor p21[WAF1/Cip1]
can repress S phase entry, resulting in CDK-mediated G1 cell cycle
arrest[45,46]. Accumulation of p21 can also function independently of p53
to trigger genomic instability by deregulating the replication licensing
machinery[47,48]. To elucidate the nature of the cell cycle arrest observed
in response to ZNF827 depletion, we investigated p21 levels. Cell cycle
arrest coincided with a striking accumulation of p21 that was inde-
pendent of p53 protein levels (Fig. 5J). Topotecan treatment resulted in
slightly elevated levels of p53, but had no effect on p21 levels. Overall,
these data demonstrate that depletion of ZNF827 causes a dramatic
induction of p21, that coincides with cell cycle arrest at the G1/S
transition.

## Discussion

The DDR is a complex and inter-related cascade of pathways that
responds to DNA damage and replication defects to orchestrate DNA

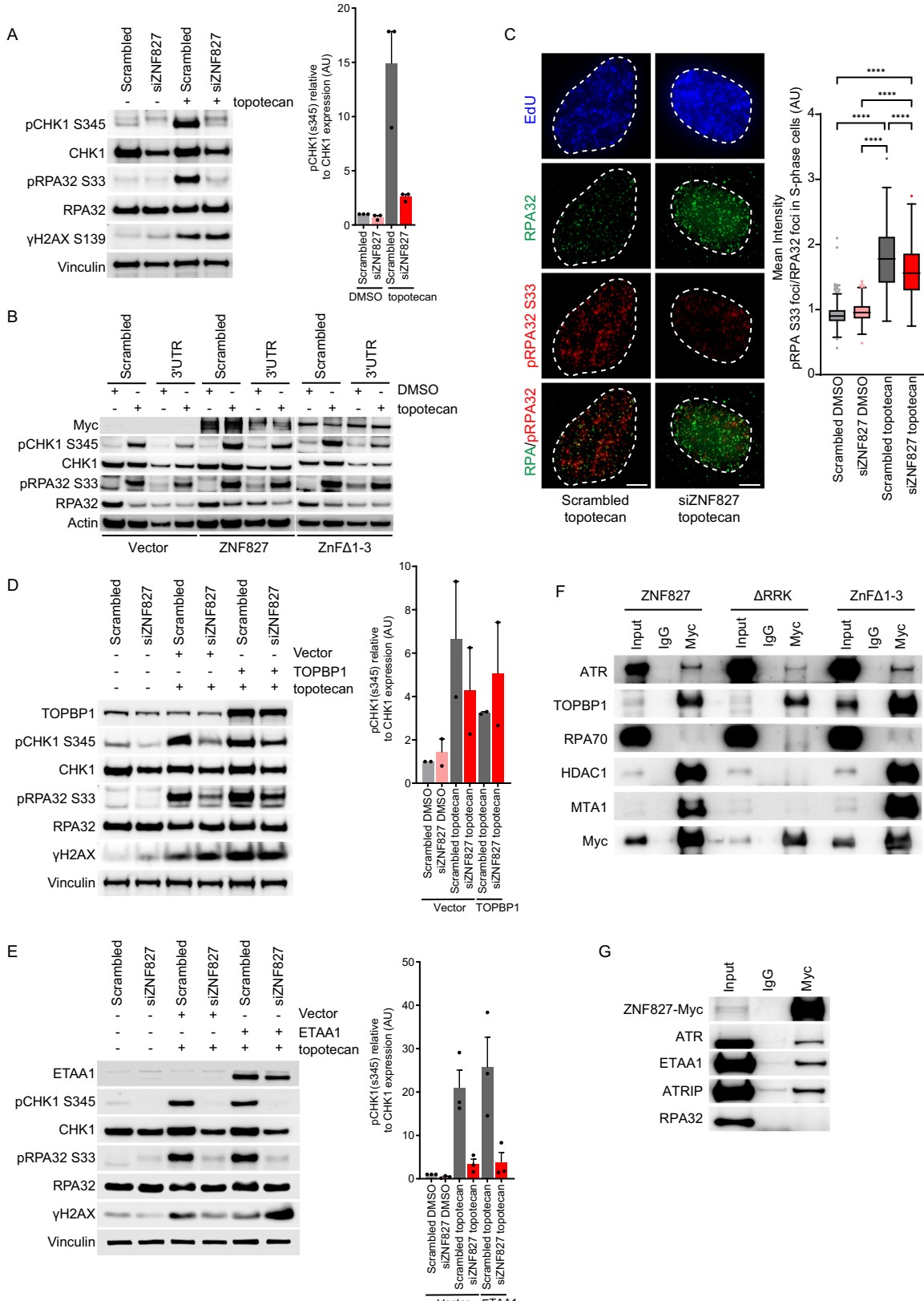

repair. While the general pathways have been mapped, the full repertoire of protein components and pathway intricacies remain to be elucidated. We identify ZNF827 as a ssDNA binding protein and component of the replication stress response. The DNA binding capabilities of other ssDNA binding proteins, such as RPA and hSSB1, are dependent on the presence of oligonucleotide/oligosaccharide-

binding (OB)-folds that facilitate ssDNA binding[49]. ZNF827 does not contain OB-folds, and instead binds to ssDNA through two clusters of C2H2 zinc finger domains, with binding relying predominantly on a central cluster of three C2H2 motifs. Precedence for C2H2 motif binding to ssDNA has recently been reported[13]. Specifically, ZPET (zinc finger protein proximal to RAD18) binds to ssDNA similarly through its

**Fig. 3 | ZNF827 is required for ATR-CHK1 kinase activation. A** Western blot analysis of pCHK1 (S345), CHK1, pRPA32 (S33), RPA32 and γH2AX S139 in U-2 OS cells following 72 h ZNF827 knockdown (left panel). Vinculin used as a loading control. Densitometry of pCHK1 (S345)/CHK1 for three biological replicates presented as mean + SEM (right panel). **B** Western blot analysis of Myc, pCHK1 (S345), CHK1, pRPA32 (S33) and RPA32 in U-2 OS cells stably expressing Myc-tagged ZNF827, ZNF827 ZnFΔ1-3, or vector control (n = 3). Endogenous ZNF827 was depleted using a ZNF827 3′ UTR siRNA. Actin used as a loading control. **C** Representative images of S-phase cells stained for EdU, RPA32 and pRPA32 (S33) following siRNA depletion of ZNF827, treatment with topotecan or DMSO and fixation in the presence of EdU. Foci identification and intensity quantitation were performed in CellProfiler. Out of three experiments, n = 373 cells scored in scrambled DMSO, n = 220 cells scored in siZNF827 DMSO, n = 408 cells scored in scrambled topotecan, and n = 258 cells scored in siZNF827 topotecan treatments. Data are presented as Tukey box plots, ****P < 0.0001, by Mann–Whitney test.

**D** Western blot analysis of TOPBP1, pCHK1 (S345), CHK1, pRPA32 (S33), RPA32 and γH2AX S139 in U-2 OS cells following 72 h ZNF827 knockdown and 48 h Myc-tagged TOPBP1 overexpression (left panel). Vinculin used as a loading control. Densitometry of pCHK1 (S345)/CHK1 for three biological replicates presented as mean + SEM (right panel). **E** Western blot analysis of ETAA1, pCHK1 (S345), CHK1, pRPA32 (S33), RPA32 and γH2AX S139 in U-2 OS cells following 72 h ZNF827 knockdown and 48 h Myc-tagged ETAA1 overexpression (left panel). Vinculin used as a loading control. Densitometry of pCHK1 (S345)/CHK1 for three biological replicates presented as mean + SEM (right panel). **F** Western blot analysis of ATR, TOPBP1, RPA70, Myc, and NuRD components HDAC1 and MTA1, following treatment with benzonase and co-immunoprecipitation (co-IP) in U-2 OS cells overexpressing Myc-tagged ZNF827, ZNF827 ΔRRK and ZNF827 ZnFΔ1-3 (n = 1). **G** Western blot analysis of Myc, ATR, ETAA1, and ATRIP following treatment with benzonase and co-immunoprecipitation (co-IP) with a Myc antibody in U-2 OS cells overexpressing Myc-tagged ZNF827 (n = 3). Source data are provided as a Source Data file.

central cluster of C2H2 motifs, and was found to slow DNA end resection, thereby restricting HR[13].

ZNF827 associates with RPA indirectly through concurrent and competitive binding to ssDNA intermediates. RPA binds ssDNA preferentially to ZNF827, consistent with RPA abundance and its high binding affinity to ssDNA. It has previously been shown that RPA occupies 20–30 nucleotides of ssDNA, forming dynamic interactions and conformations through its modular domain architecture[50,51]. RPA plays a dual role in protecting ssDNA from nucleolytic degradation and hairpin formation, as well as coordinating the sequential assembly and disassembly of DNA replication and repair proteins on ssDNA[9,51,52]. Our data indicate that ZNF827 plays an independent, yet complementary, role to RPA at ssDNA. We speculate that ZNF827 may recognize and respond to distinct ssDNA species, or may elicit stoichiometric changes that confer particular binding interactions or cellular outcomes. Our data show that ZNF827 localizes to RPA and PCNA, suggesting that ZNF827 may contribute to normal DNA replication by processing ssDNA intermediates either ahead or behind the progressing replication fork to expedite fork progression.

ZNF827 accumulates at sites of replication-associated DNA damage. This aligns with the increased prevalence of ssDNA at these sites. We demonstrate that ZNF827 interacts with both the ATR activators TOPBP1 and ETAA1, as well as ATRIP, and ATR itself. These interactions are independent of both ssDNA binding and the presence of NuRD. Depletion of ZNF827 suppresses activation of the ATR-CHK1 kinase pathway, despite the persistence of DNA damage. The ATR activation defect can be partially rescued by overexpression of both TOPBP1 and ZNF827 and is dependent on the ssDNA binding capability of ZNF827. Together, these data support a role for ZNF827 in facilitating the handover from ATR-mediated DNA damage sensing to HR pathway engagement. Consistent with this rationale, depletion of ZNF827 suppresses HR-mediated DNA repair. These observations are particularly striking in response to induced replication stress-associated DNA damage, which confers robust ATR activation and HR pathway engagement that is substantially subdued in the absence of ZNF827.

We further demonstrate that depletion of ZNF827 diminishes the frequency of second origins and reduces the intensity of EdU staining in S phase cells, indicative of replication inhibition in response to ZNF827 depletion. This coincides with a striking accumulation of p21 and cell cycle arrest at the G1/S transition. Specifically, ZNF827 depleted cells display severe cell cycle defects, including mitotic bypass, cell division in G1, and failure to commit to S phase resulting in reversion back to G1. Failure of commitment to S phase can also result in transition out of the cell cycle to a reversible non-dividing quiescent state (G0)[46]. Quiescence is characterised by replication shut-down and an accumulation of p21, and can be induced in response to replication stress to temporarily isolate cells from aspects of the DDR[46,53–55].

Both G1 and quiescent cells express CDT1, and therefore cannot be distinguished by FUCCI. This implicates quiescence as a potential outcome of ZNF827 depletion that requires further delineation.

We have previously reported that ZNF827 recruits the NuRD complex to telomeres engaged in HR-mediated telomere maintenance through the ALT pathway[33]. This highlights the parallels between the HR pathways engaged at ALT telomeres and genomic HR-mediated repair and introduces a protein previously thought to act specifically at telomeres to the genomic DDR repertoire. We demonstrate that ZNF827 nuclear localization, binding to ssDNA, and the interaction between ZNF827 and TOPBP1, are independent of ZNF827 binding to NuRD. Nevertheless, ZNF827 is required for BRCA1 foci formation and the recruitment of BRCA1 to sites of replication stress. These data suggest that ZNF827 functions to bridge the genomic response to replication stress to the engagement of HR, potentially through NuRD-mediated recruitment of HR factors. ZNF827 localisation to ALT telomeres is likely attributed to the abundant telomeric ssDNA species that are generated during the ALT mechanism. However, subsequent engagement of HR appears to involve NuRD complex functionality. This is supported by the involvement of NuRD components in DNA repair and the reported role of the zinc finger protein ZMYND8 in recruiting NuRD to DNA damage sites to promote HR-mediated repair[56–60].

In summary, our data identify ZNF827 as a component of the replication stress response that plays a fundamental role in maintaining genome stability. ZNF827 binds directly to ssDNA and accumulates at sites of replicative damage, where it activates the ATR kinase pathway to promote HR-mediated repair. Depletion of ZNF827 inhibits replication initiation and fork progression, and results in an accumulation of DNA damage, p21 induction, and cell cycle arrest at the G1/S transition. ZNF827 depletion sensitizes cancer cells to treatment with the chemotherapeutic topotecan. These effects implicate ZNF827 as a molecular target for the development of cancer therapeutics directed at ATR pathway inhibition.

## Methods
### Cell culture and cell lines
The cell lines U-2 OS, HT1080, IIICF/c, HT10806TG and HEK-293T were cultured in Dulbecco's modified Eagle's medium (DMEM) supplemented with 10% (v/v) fetal bovine serum (FBS) in a humidified incubator at 37 °C with 10% $CO_2$. Cell lines were authenticated by 16-locus short-tandem-repeat (STR) profiling and tested for mycoplasma contamination by CellBank Australia (Children's Medical Research Institute).

### Vectors and gene expression
pCMV6-Entry Myc-DDK-tagged ZNF827, pCMV6-AC-GFP-ZNF827 and pCMV6-Entry empty vector were obtained from OriGene Technologies. pHTN HaloTag® CMV-neo empty vector was

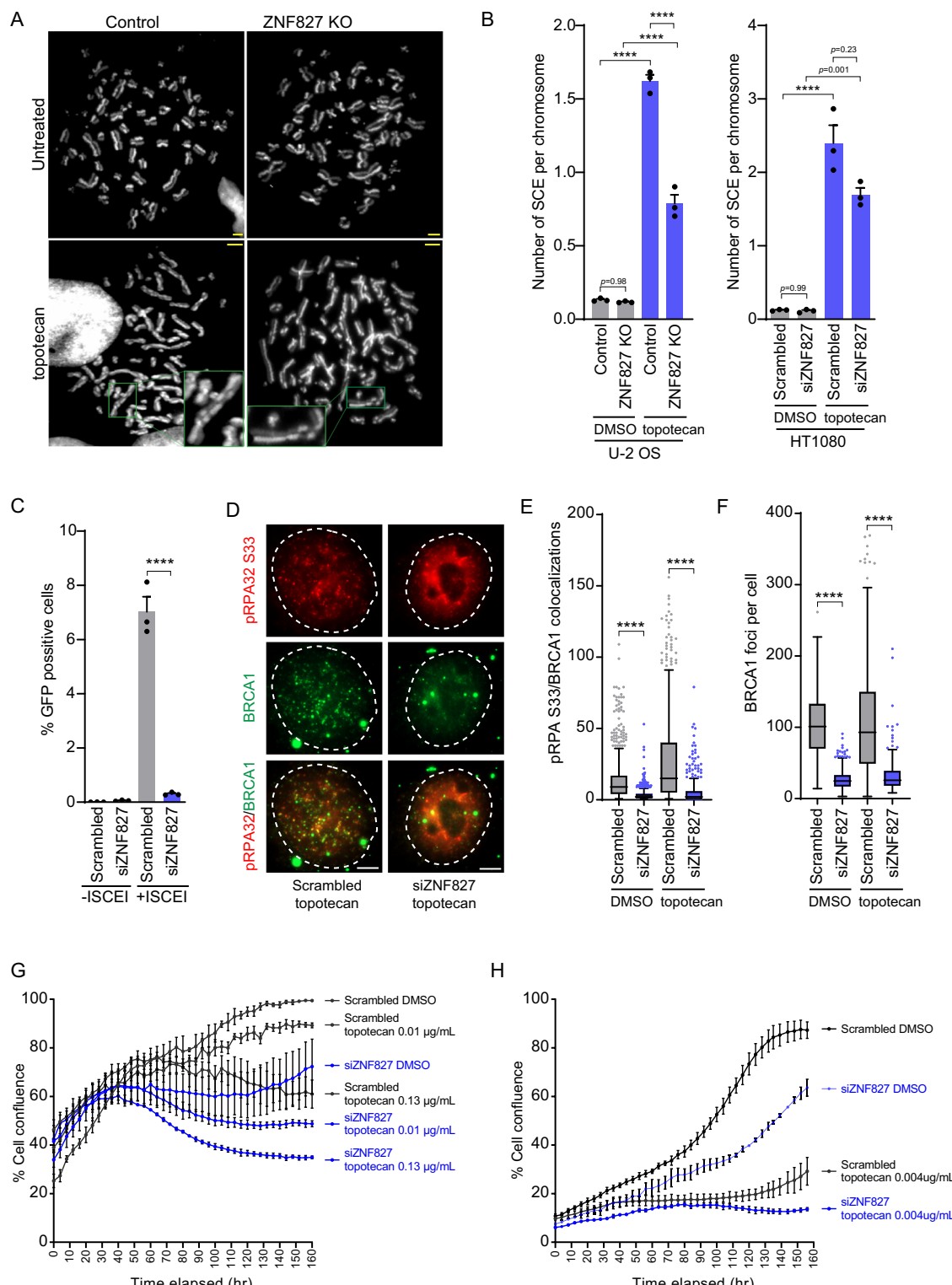

obtained from Promega. ZNF827 was sub-cloned into the pHTN HaloTag® CMV-neo backbone by restriction enzyme cloning at SgfI and NotI sites. pCMV6 Myc-DDK-tagged ZNF827 ZnF1-3 deleted (ZnFΔ1-3) and pHTN HaloTag® CMV-neo ZNF827 ZnF1-3 deleted (ZnFΔ1-3) mutant constructs were synthesized by Genscript. The ZnF4-9 deleted (ZnFΔ4-9) mutant was generated by restriction enzyme excision of ZnF4-9, and subcloning of a fill-in gene block purchased from Integrated DNA Technologies. Similarly, the truncated mutant was generated by restriction enzyme excision of ZnF4-9 from the ZnF1-3 deleted mutant, and subcloning of a fill-in

gene block. The ΔRRK mutant was generated previously, using site-directed mutagenesis (Agilent Technologies)[33]. pCMV6-AC-GFP RPA32, pCMV6-Entry Myc-DDK-tagged ETAA1 and pCMV6-Entry Myc-DDK-tagged TOPBP1 were obtained from Origene Technologies.

Transient expression of plasmid constructs was performed using FuGENE-6 transfection reagent (Promega) according to the manufacturer's instructions. Overexpression was assayed using western blot analysis, and cells were harvested 48 h after transient transfection with expression constructs unless stated otherwise.

**Fig. 4 | ZNF827 promotes HR-directed DNA repair. A** Representative images of sister-chromatid exchange (SCE) events in U-2 OS and U-2 OS ZNF827 CRISPR KO cells incubated with topotecan or DMSO. Green boxes indicate sites of exchange events. Scale bar represents 5 μm. **B** Quantitative data of SCE events in U-2 OS and U-2 OS ZNF827 CRISPR KO cells incubated with topotecan or DMSO (left panel), and in HT1080 cells following 72 h ZNF827 depletion, including treatment with topotecan or DMSO (right panel). Data presented as mean + SEM from 800–1000 chromosomes per replicate from three biological replicates; ****$P < 0.0001$ by one-way ANOVA **C** HR reporter assay in U-2 OS cells stably expressing pDR-GFP. Quantitation of GFP-positive cells detected by flow cytometry. Data presented as mean + SEM from three independent experiments; ****$P < 0.0001$ by two-tailed $t$-test. **D** Representative images of BRCA1 (green) and pRPA32 (S33) (red) foci in HT1080 cells. Scale bar represents 5 μm. Quantitation of **E** BRCA1 colocalizations with pRPA32 (S33) per nucleus, and **F** BRCA1 foci per nucleus in HT1080 cells. Out

of three experiments, $n = 930$ cells scored in scrambled DMSO, $n = 694$ cells scored in siZNF827 DMSO, $n = 1054$ cells scored in scrambled topotecan, and $n = 458$ cells scored in siZNF827 topotecan using CellProfiler. At least 25% overlap criterion was applied for colocalization count. Data are representative of three independent experiments and presented as Tukey box plots; ****$P < 0.0001$ by Kruskal–Wallis tests. **G** Growth curves by cell confluency obtained from IncuCyte® live cell assays of U-2 OS cells following ZNF827 depletion. Topotecan or DMSO were added at 24 h post-siRNA knockdown and cells imaged every 4 h up to 160 h. Data presented as mean ± SEM from two biological replicates. **H** Growth curves by cell confluency obtained from IncuCyte® live cell assays of HT1080 cells following ZNF827 depletion. Topotecan or DMSO were added at 24 h post-siRNA knockdown and cells imaged every 4 h up to 160 h. Data presented as mean ± SEM from two biological replicates. Source data are provided as a Source Data file.

Stable expression of ZNF827 and ZNF827 ZnF1-3 was achieved by lentiviral transduction. Specifically, ZNF827 and ZNF827 ZnF1-3 were sub-cloned into pLenti-C-Myc-DDK-IRES-puro (Origene) by restriction enzyme digestion at SgfI and NotI sites, followed by ligation, purification, and sequence verification. For transduction, cells were treated with 8 μg/ml polybrene (Merck) concurrently with lentiviral particles containing plasmid constructs for 48 h. Cells were selected in 2 μg/ml puromycin (U-2 OS), and overexpression was confirmed by western blot.

### Mass spectrometry
ZNF827 pull-downs were performed in HEK 293 T cells overexpressing HaloTag-ZNF827 ($n = 1$) or empty vector control ($n = 1$) using the HaloTag® Protein Pull-Down and Labelling System (Promega) according to the manufacturer's protocol. Pull-downs were subjected to mass spectrometry (MS) for purity validation. Sample preparation and MS were performed as described previously[61]. Briefly, empty vector controls and ZNF827 HaloTag pull-downs from HEK 293 T were digested with trypsin and prepared for LC–MS/MS using the optimised Accelerated Barocycler Lysis and Extraction (ABLE) method[61], where following sample reduction and alkylation, digestion was carried out in the Barocycler using 30 cycles of 50 s at 45 kpsi and 10 s at atmospheric pressure, at 70 °C. Samples were acidified with formic acid, then centrifuged (15 min 18,000 g). The supernatant was transferred to a new tube and evaporated in a speedyvac to dryness. Samples were resuspended in 0.1% (v/v) formic acid, and the concentration determined using A280 nm with a Implen nanophotometer N60.

Peptide spectra were acquired with the IDA (information-dependent acquisition) LC–MS/MS method as described[61], using the Triple TOF6600 system (SCIEX) equipped with a DuoSpray source and 50 μm internal diameter electrode and controlled by Analyst1.7.1 software. The following parameters were used: 5500 V ion spray voltage; 25 nitrogen curtain gas; 100 °C TEM, 20 source gas 1, 20 source gas 2. The 90 min information-dependent acquisition (IDA) consisted of a survey scan of 200 ms (TOF-MS) in the range 350 – 1250 m/z to collect the MS1 spectra and the top 40 precursor ions with charge states from +2 to +5 were selected for subsequent fragmentation with an accumulation time of 50 ms per MS/MS experiment for a total cycle time of 2.3 s, and MS/MS spectra were acquired in the range 100 – 2000 m/z.

Spectra data were searched using Mascot Server 2.6.2 against the SWISSPROT databases (release 2016_10) with the following parameters - fixed modification of Cys-CAM; variable modification of oxidation (M) and deamidation (N,Q); enzyme trypsin; and max missed cleavage of 2. The peptide list identified from the ZNF827 pull-down were compared to the list from the empty vector control qualitatively to exclude common peptides, identifying peptides specific to the ZNF827 pull-down. No other protein having a score > 100 and coverage greater than 0.15 was used to confirm ZNF827 purity. The mass spectrometry proteomics data have been deposited to the ProteomeXchange Consortium via the PRIDE partner repository with the dataset identifier PXD050191.

### RNA interference
The following siRNAs were designed and synthesized by Life Technologies: ZNF827 (Stealth RNAi siRNA ZNF827HSS135819), Stealth RNAi™ Negative Control Med GC Duplex #2 (12935-112), RPA32 (Silencer Select RNAi siRNA s12131), Silencer Select RNAi siRNA Negative Control Med GC Duplex #2. Cells were seeded at 30-40% confluency, transfected using Lipofectamine RNAiMAX transfection reagent (ThermoFisher Scientific) with an siRNA concentration of 20 μM according to the manufacturer's instructions, and harvested for analysis 72 h post-transfection. Knockdown was validated using qRT-PCR or western blot analysis.

### ZNF827 CRISPR knockout
ZNF827 knockout by CRISPR/Cas9 genome editing was performed by the Vector and Genome Engineering Facility (VGEF) at Children's Medical Research Institute in U-2 OS, HT1080 and HEK 293 T cells. One complete ZNF827 knockout clone was obtained from U-2 OS, while no viable clones were obtained from HT1080 and HEK 293 T.

### Quantitative reverse transcription PCR (qRT-PCR)
Knockdown of ZNF827 was verified 72 h post-siRNA transfection by qRT-PCR analysis, as described previously with minor modifications[33]. Briefly, total RNA was homogenized with the QIAshredder mini kit (Qiagen), extracted with the RNeasy Mini Kit (Qiagen), and quantified by spectrophotometry. cDNA synthesis was performed using SuperScript III reverse transcriptase (Invitrogen). PCR reactions for ZNF827 and GAPDH were conducted with the following forward (F) and reverse (R) primers with Platinum™ SYBR™ Green qPCR SuperMix (ThermoFisher Scientific): ZNF827 F, GGCTCAACTCAGGACAGTGG; ZNF827 R, CCGGCACTTGTACTCCATCTT; GAPDH F, ACCCACTCCTCCACCTTTG; GAPDH R, CTCTTGTGCTCTTGCTGGG. Reverse-transcriptase negative controls and water controls were included. PCR conditions were 95 °C for 10 min, 40 cycles of 95 °C for 15 s, 60 °C for 60 s and 72 °C for 30 s, followed by melt curve analysis. Matched PCR efficiencies for the primer sets were confirmed by standard curve comparison. Analysis was performed with LightCycler® 96 system software, using the ΔΔCt method with GAPDH as the reference gene.

### Western blot analysis
Western blot analysis was conducted as described previously[34]. Uncropped and unprocessed scans of blots are provided in the Source Data file.

### Antibodies
A complete list of all primary antibodies (with manufacturers, catalogue numbers, and dilution factors) used throughout this study can be found in Supplementary Table 2. Fluorophore-conjugated secondary antibodies (Life Technologies) were used for indirect immunofluorescence. Polyclonal goat anti-mouse, rabbit, and goat immunoglobulin conjugated to horseradish peroxidase were used for western blot analysis (DAKO).

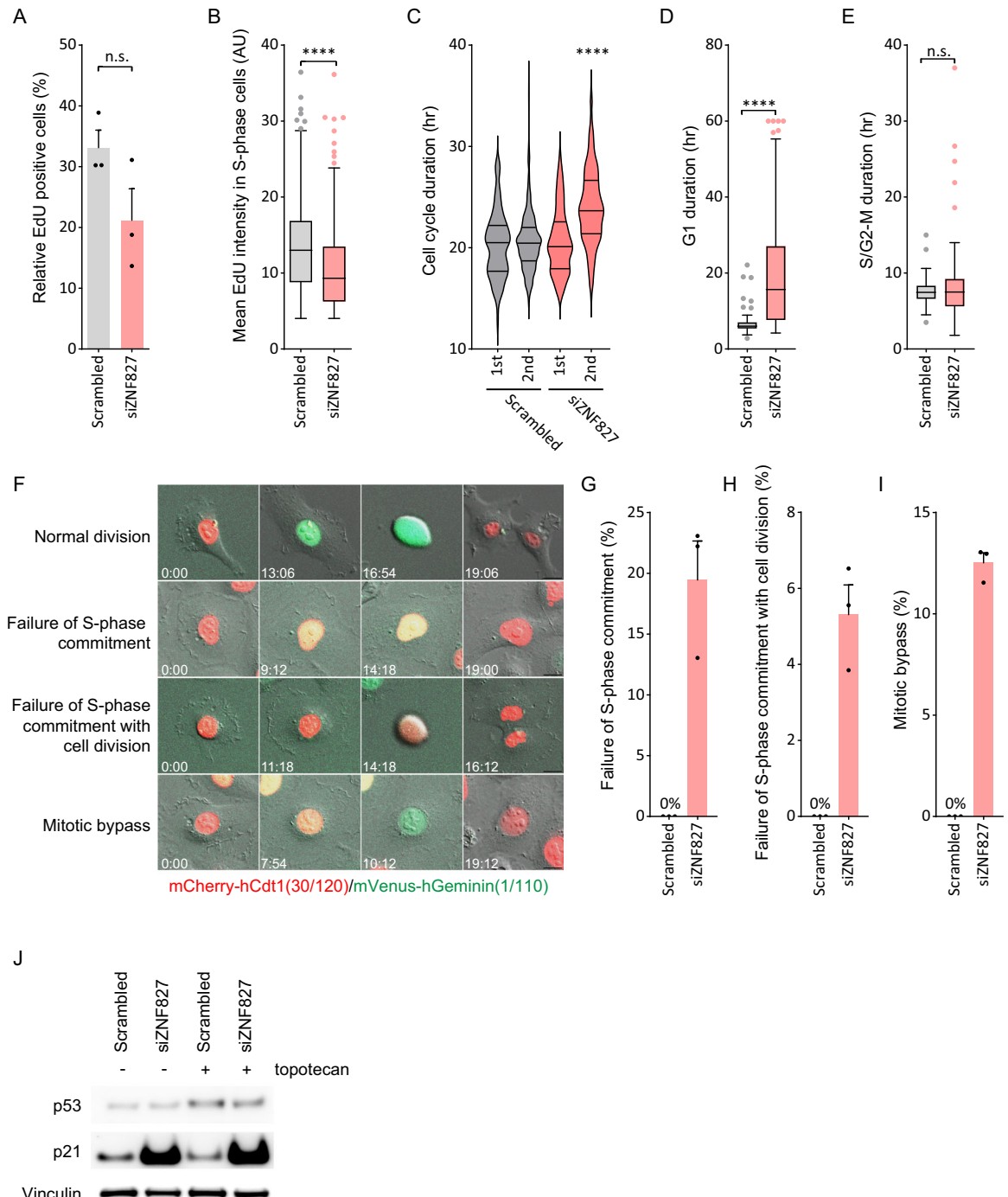

**Fig. 5 | ZNF827 depletion results in cell cycle dysregulation. A** Quantitation of EdU-positive U-2 OS cells 72 h post ZNF827 depletion. Data presented as mean + SEM from at least 500 cells across three independent experiments. n.s. non-significant ($p = 0.1337$) by two-tailed unpaired Welch's $t$-test. **B** Quantitation of mean EdU intensity in EdU-positive S-phase cells. Data presented as Tukey box plots from $n = 975$ cells in scrambled and $n = 546$ cells in siZNF827 treated U-2 OS cells across three independent experiments. ****$P < 0.0001$ by two-tailed unpaired Mann–Whitney test. **C** Cell cycle duration analysis of U-2 OS live cell imaging data from 6 h post ZNF827 siRNA transfection. Cell cycle duration was assayed in $n = 98$ and $n = 88$ scrambled and siZNF827 cells, respectively. Data presented as Violin plots; ****$P < 0.0001$ by Mann–Whitney test. Quantitation of live cell imaging in HT1080 FUCCI cells following ZNF827 depletion showing **D** G1 duration, **E** S/G2-M duration. Out of three experiments, $n = 74$ scrambled and $n = 104$ siZNF827 cells were assayed for G1 duration; $n = 74$ scrambled and $n = 59$ siZNF827 cells were assayed for S/G2-M duration. Data presented as Tukey box plots, ****$P < 0.0001$, n.s. = non-significant ($P = 0.91$) by two-tailed unpaired Mann–Whitney test. **F** Representative images from live cell imaging of HT1080 FUCCI cells showing cell division in G1. Time is shown as hours: minutes relative to the first image of the series. Quantitation of live cell imaging in HT1080 FUCCI cells following ZNF827 depletion showing **G** proportion of cells that transition from G1 to S to G1 (failure of S phase commitment), **H** proportion of cells that divide in G1 (failure of S phase commitment with cell division), and **I** proportion of cells that undergo mitotic bypass. Data presented as mean + SEM from 40 cells per replicate from three biological replicates. n.s. non-significant, ****$P < 0.0001$ by two-tailed $t$-test. **J** Western blot analysis of p21 and p53 in U-2 OS cells following 72 h ZNF827 depletion, including topotecan treatment ($n = 3$). Vinculin used as a loading control. Source data are provided as a Source Data file.

## Protein purification

ZNF827 full length, ZNF827 ZnF1-3 deleted (ZnFΔ1-3), ZNF827 ZnF4-9 deleted (ZnFΔ4-9) and ZNF827 truncated recombinant proteins were transiently overexpressed in HEK 293 T cells and purified using the HaloTag® Protein Detection and Purification System (Promega) according to the manufacturer's instructions. Briefly, cell pellets were resuspended homogenously in Mammalian Lysis Buffer (Promega) supplemented with $1\times$ Protease Inhibitor Cocktail (Promega) to a concentration of $2-6\times10^7$ cells/mL, and incubated at 4 °C for 1 h on a rotator. Cell lysates were centrifuged at $10,000\times g$ for 30 min at 4 °C to collect the supernatant, which was then diluted 1:3 by adding HaloTag® Protein Purification Buffer. 180 μL/mL lysis buffer of HaloLink™ Resin slurry was equilibrated by five washes, each for 5 min, in 5 mL of HaloTag® Purification Buffer followed by removal of supernatant after centrifugation at $1500\times g$ for 5 min between washes. Cell lysates were added to the equilibrated resin, mixed well by inverting the tubes, and incubated on an rotator at 4 °C overnight. The following day, samples were centrifuged at $1500\times g$ for 5 min. Supernatants were transferred to another tube as flowthrough fraction for binding efficiency analysis as required. The resin bound with proteins was washed with 5 mL HaloTag® Protein Purification Buffer on a rotator for $3\times10$ min followed by removal of supernatant after centrifugation at $1500\times g$ for 5 min between washes. For protein elution, HaloTEV Protease cleavage solution (9 μL of HaloTEV Protease in 291 μL HaloTag® Protein Purification Buffer per 900 μL resin slurry) was added to the settled resin, mixed well and incubated on a rotator at 4 °C overnight. The following day, the supernatant (eluate) was collected by centrifugation at $1500\times g$ for 5 min. To remove residual resin, the eluate was transferred to a spin column in a 1.5 mL protein Lobind™ microcentrifuge tube (Eppendorf) and collected by centrifugation at $10,000\times g$ for 15 s. Eluates were mixed with 10% glycerol to maintain protein stability, aliquoted and stored at −80 °C until use. Purified proteins were quantitated by BCA (bicinchoninic acid) assays, and analysed by Coomassie gels, western blots and mass spectrometry.

Replication protein A (RPA) was purified as described previously[62]. Briefly, 2 L of BL21(DE3) containing plasmid pET11d-tRPA was induced with IPTG to final concentration 0.6 mM when the culture reached an $OD_{600nm}$ of 0.5. After 3 h, cells were collected by centrifugation and lysed in HI buffer (30 mM HEPES pH 7.8, 0.25 mM EDTA, 0.25% Myo-inositol, 1 mm DTT, 0.01% IGEPAL CA630) supplemented with 50 mM KCl and bacterial protease inhibitor cocktail (Sigma Life Sciences). Cells were disrupted by passing them through an Emulsiflex C-5 high-pressure homogenizer. Cell debris was removed by centrifugation and the lysate applied to an Affigel Blue column (Biorad). RPA was eluted with HI buffer containing 1.5 M NaSCN. Fractions containing RPA were pooled and dialysed against HI buffer containing 50 mM KCl. The soluble RPA complex was loaded onto Hydroxyapatite (BioRad) column, washed with HI buffer containing 80 mM KCl and RPA eluted with HI buffer with 800 mM KCl. RPA-containing fractions were pooled and diluted 1:1 with HI buffer with 50 mM KCl and loaded onto a MonoQ PE (GE) column. RPA was eluted with a gradient of HI buffer from 50–800 mM KCl. Peak fractions were pooled and stored at −80 °C.

## Co-immunoprecipitation

Co-immunoprecipitation experiments were conducted as described previously[34].

## Electrophoretic mobility shift assay (EMSA)

Purified protein samples of equal amounts were incubated in binding buffer (20 mM HEPES-KOH, pH7.9, 100 mM KCl, 0.8 mM ZnCl₂, 0.2 mM EDTA, 5% glycerol, 0.5 mM DTT and 0.5 mM PMSF) with 6.25 μg/mL poly (dI:dC) and 50 ng/μL BSA on ice for 20 min. -0.5 nM γ-$^{32}$P labelled single-stranded (ss) DNA oligonucleotide pentaprobe: 5′-CGCTCTATTCTACTGTCCTGTGCATTCAATCGTTGAGTTCGATCTAGT

CTCGTCT AACCCTCCCCTGCTCCGCTGGTCTGGCCTCGCCTATCCTA CCCAT-3′ or double-stranded (ds) oligonucleotide probe annealed from 5′-CGCTCTATTCTACTGTCCTGTG CATTCAATCGTTGAGTTC-GATCTAGTCTCGTCTAACCCTCCCCTGCTCCGCTGGTCTGGCCTCGC CTATCCTACCCAT-3′ and 5′-ATGGGTAGGATAGGCGAGGCCAGA CCA GCGGAGCAGGGGAGGGTTAGACGAGACTAGATCGAACTCAACGATTG AATG CACAGGACAGTAGAATAGAGCG-3′[35] were added to samples and incubated on ice for another 30 min. Annealed dsDNA probes were digested by ExoVII to remove any residual single-stranded regions. For experiments with sequential additions of proteins, the first protein was incubated with γ-$^{32}$P labelled DNA probes for 15 min prior to the addition of the second protein. Binding reactions were loaded onto an 8% native acrylamide/bisacrylamide (19:1, Biorad) gel pre-run at 100 V in $0.5\times$ TB buffer (89 mM Tris, 89 mM boric acid) for 20 min. Gels were electrophoresed at 180 V for 120 min at 4 °C, dried at 65 °C for 45 min, and then exposed to a Phosphor screen (GE Life Sciences) overnight. Gel images were obtained by scanning the screens with a Typhoon FLA9500 Imager (GE Life Sciences). DNA oligonucleotide probes were end labelled with γ-$^{32}$P using T4 polynucleotide kinase (New England Biolabs) according to the manufacturer's instructions.

## Indirect immunofluorescence (IF)

Cells were cultured on sterile Alcian blue-stained-glass coverslips (Marienfeld-Superior) in 6 or 12-well tissue culture plates (Corning). Indirect immunofluorescence was performed on interphase nuclei as described previously[63]. To detect ssDNA, cells were pre-labelled with 10 μg/mL BrdU for 24 h, as described[64]. Primary and secondary antibodies used in IF can be found in Supplementary Table 2.

## Sister-chromatid exchange (SCE)

20–25% confluent cells were cultured in fresh media supplemented with 7.5 μM BrdU and 2.5 μM BrdC (BrdU:BrdC 3:1 ratio; Sigma-Aldrich) for 32–48 h depending on the mitotic index of the cell line. Cell cultures were treated with 100 ng/mL colcemid for the last 4 h of incubation to accumulate mitotic cells. Cells were harvested by trypsinization and centrifugation and then incubated in hypotonic buffer for 10 min at 37 °C. Swollen cells were fixed by gradually adding 1 mL of fresh ice-cold fixative (methanol/acetic acid 3:1), mixing by inversion and incubating on ice for 5 min. Cells were then collected by centrifugation at $1500\times g$ for 8 min. 10 mL ice-cold fixative was added to resuspend the cells followed by 5 min incubation on ice and centrifugation at $1500\times g$ for 8 min. This fixing step was repeated another two times. Fixed cells were then resuspended in 500–1000 μL of ice-cold fixative, and dropped onto clean, dry, microscope slides (HDS Surefrost, 50–100 μL cell solution per slide). To drop chromosomes, a clean dry slide was held over a 75 °C water bath, and the cell solution was dropped from a pipette onto the slide, which was quickly flipped and held close to the surface of water bath for 5 s. Slides were left to dry for 2 to 3 days and then treated with 100 μg/ml DNase-free RNase A (Sigma) in $2\times$ SSC for 30 min at 37 °C, rinsed in PBS, and postfixed in 4% formaldehyde in PBS at room temperature for 10 min. Following a quick rinse in deionized water, slides were dehydrated in a graded ethanol series (70% for 3 min, 90% for 3 min, and 100% for 3 min) and allowed to airdry. Slides were then stained in 0.5 μg/mL Hoechst 33258 (Sigma−Aldrich) in $2\times$ SSC for 15 min at room temperature, rinsed in dH₂O, and air-dried. Slides were then flooded with 200 μL $2\times$ SSC and exposed to long-wave (~365 nm) UV light (Stratalinker 1800 UV irradiator; Agilent Technologies) for 45 min. The BrdU/BrdC-substituted DNA strands were then digested in 10 U/μL Exonuclease III solution (New England Biolabs) in the supplied buffer at 37 °C for 30 min. After a quick rinse in deionized water, slides were incubated with 50 ng/mL DAPI in PBS for 15 min, washed twice in PBST for 5 min, rinsed in deionized water, and airdried. Airdried slides were mounted in Prolong Gold Antifade (Invitrogen) and stored at 4 °C until microscope analysis. Slides were imaged by automation on the MetaSystems

Metafer Scanning Platform (Carl Zeiss) microscope, and analysed manually using Isis software (Metasystems).

## Flow cytometry

Flow cytometry was performed as described previously[41]. Briefly, ethanol-fixed single-cell suspensions were stained with propidium iodide (PI). Cells were analyzed by BD FACSCanto Flow Cytometry (BD Biosciences) using an air-cooled 488 nm argon laser to excite PI. Forward scatter and side scatter of each cell were recorded. Doublets identified as cells with 4 N DNA content and increasing pulse width were eliminated. The percentage of cells in each cell cycle stage was calculated using FlowJo v5 software (FlowJo).

## Live cell assays

Cells were seeded in glass bottom 12-well plates (MakTek Corporation) the day before imaging at 30% confluency, to ensure that cells were actively cycling for the duration of the experiment. Media was replaced with phenol red-free DMEM (Gibco) supplemented with 10% FCS and drug treatment immediately before transferring the plate to the microscope chamber. Imaging was performed on the Cell Observer Widefield Microscope (Zeiss) using a 20× objective at 37 °C with 10% $CO_2$ in a XLmulti S1 full enclosure chamber. Cells were incubated in the XLmulti S1 full enclosure chamber for 2 h prior to imaging. Three to four positions per well were captured with AxioCam (Zeiss) 506 Mono using ZEN software (Carl Zeiss) with images taken every 6 min for 60 h. The number of nuclei, interphase duration, S/G2 duration and mitotic duration (defined by duration between initial rounding up of cell to completion of cytokinesis) were recorded. More than 15 cells were analysed at each position for each condition. Live cell assays for GFP-RPA (Origene Technologies) and TMRdirect-Halotag-ZNF827 (Promega), GFP-ZNF827, and chromobody-RFP-PCNA (ChromoTek) were performed on the Cell Observer SD 48 h post-transfection using a 40× objective at 37 °C with 10% $CO_2$, with images taken every 5 min.

## Cell proliferation assay using IncuCyte®

Cells were seeded at 20–30% in a 96-well black plate with clear bottom (3603, Corning) and left to adhere at 37 °C with 10% $CO_2$ for at least 3 h. Media was replaced with phenol red-free media containing topotecan at a titrating range of concentrations, and placed into the IncuCyte®. The plate was imaged every 4 h for 120 h in the phase channels. Image analysis was performed with the IncuCyte® Zoom software. Cell growth was measured as % confluency over time.

## Single-molecule analysis of replicated DNA (SMARD)

SMARD analysis was conducted as described previously[63]. Briefly, labelled cells were embedded in agarose plugs and subjected to proteinase K digestion. Molecular combing was performed using a constant stretching factor on vinyl silane-coated coverslips. Fibre quality and integrity was verified by YoYo-1 staining. Coverslips were denatured and fixed, and halogenated nucleotides detected by immunofluorescence. For replication rate analysis, only elongating DNA tracts were measured.

## Homologous recombination (HR) reporter assay

HR reporter assays were performed as described previously, with minor modifications[43,65]. Briefly, U-2 OS cells were transfected with HR reporter plasmid DR-GFP using FuGENE-6 transfection reagent (Promega), and selected for stable integration using puromycin at 4 μg/mL 48 h post-transfection. Puromycin-resistant clones were selected, expanded, and screened for baseline reporting activity by measuring GFP expression using flow cytometry following 48 h ISCE-I transient expression. Clones with ~6% baseline reporting events were used for HR reporter assays following ZNF827 siRNA knockdown. GFP-positive cells were analysed by BD FACS Canto II (BD Biosciences) and quantitated by FlowJo v10 software (FlowJo).

## EdU labelling of cells

Cells were cultured on sterile Alcian blue-stained-glass coverslips (Sigma–Aldrich) in 6 well tissue culture plates (Corning) using Dulbecco's Modified Eagle Medium (DMEM) (Gibco). Following 71 h siRNA transfection and 1 h topotecan treatment, cells were washed once with DMEM media and allowed to recover in fresh media containing no drug treatment. After 30 min recovery, 5-ethynyl-2′-deoxyuridine (EdU) was added to the solution to a final concentration of 10 μM. Cells were incubated for a further 30 min before harvesting at 72 h post-transfection. To harvest, cells were washed briefly with PBS, fixed in 4% formaldehyde/PBS solution for 10 min, and then permeabilised with KCM buffer (120 mM KCl, 20 mM NaCl, 10 mM Tris pH 7.5, 0.1% Triton X-100) for 10 min. Coverslips were treated with EdU Click-it reaction kit (Invitrogen) as per manufacturer's protocol. Once labelled, coverslips were incubated with 50 ng/mL DAPI in deionised water for 15 min, rinsed in deionised water, and airdried. Airdried slides were mounted onto slides using ProLong Gold Antifade Mountant (Invitrogen) and stored at 4 °C until microscope analysis. Microscopy images were acquired on a Zeiss Axio Imager microscope using 40× objective and quantitated using Cell Profiler analysis software. EdU intensity per nucleus was quantitated and S phase/EdU-positive cells defined as the population of cells with EdU intensity above 400 (arbitrary units).

## ZNF827-ssDNA immunoprecipitation dotblot

Purified ZNF827 was incubated with 0.2 pmole ss or ds oligonucleotides, as described above, for 2 h at 4 °C. ZNF827 or IgG antibodies were preincubated with Dynabeads Protein G, as per manufacturer's instructions (Life Technologies). Dynabeads conjugated to ZNF827 or IgG antibodies were added to the ZNF827 and DNA reaction at 1 μg/100 μL binding reaction, followed by incubation with rotation overnight at 4 °C. Dynabeads were then washed twice in DNA binding buffer (20 mM HEPES-KOH, pH7.9, 100 mM KCl, 0.8 mM $ZnCl_2$, 0.2 mM EDTA, 5% glycerol, 0.5 mM DTT and 0.5 mM PMSF) and separated on a magnetic rack (Invitrogen). Dynabeads were then resuspended in elution buffer (50 mM $NaHCO_3$, 1% SDS), digested with Proteinase K at 55 °C for 1 h, and heated at 70 °C for 10 min. DNA eluates were denatured in 0.46 M sodium hydroxide at 95 °C for 5 min, cooled on ice and then dot-blotted onto a Biodyne B 0.45 μm nylon membrane (Pall). The membrane was airdried for 30 min, and DNA UV-cross-linked to the membrane at 240 mJ and 254 nM using a Stratalinker, and prehybridized in PerfectHyb Plus hybridization buffer (Sigma–Aldrich) for 1 h at 55 °C. The membrane was then incubated with radiolabelled DNA probe overnight at 55 °C, followed by three 10 min washes in 2 × SSC at room temperature and air drying before being exposed to a Phosphor screen overnight. The Phosphor screen was imaged using a Typhoon FLA9500 Imager (GE Healthcare Life Sciences).

## Reporting summary

Further information on research design is available in the Nature Portfolio Reporting Summary linked to this article.

# Data availability

The mass spectrometry proteomics data have been deposited to the ProteomeXchange Consortium via the PRIDE partner repository with the dataset identifier PXD050191. All remaining data supporting the findings of this study are available within the paper and its Supplementary Information. Source data are provided with this paper.

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

## Acknowledgements

The authors acknowledge the CMRI ACRF Telomere Analysis Centre supported by the Australian Cancer Research Foundation and the CMRI Vector and Genome Engineering Facility. The authors acknowledge support from Luminesce Alliance. The Alliance is comprised of five partners, Sydney Children's Hospitals Network, CMRI, Children's Cancer Institute, University of Sydney, and University of New South Wales. We thank Sadia Mahboob and Peter G. Hains for assistance with the mass spectrometry, and Pragathi Masamsetti for assistance with cell cycle analysis. HAP is supported by the National Health and Medical Research Council of Australia (1187606, 1162886), the Medical Research Future Fund (2007488), and Cancer Council New South Wales (RG 16-09). APS was supported by Cancer Institute NSW (ECF171269).

## Author contributions

H.A.P. and S.F.Y. conceived the study. H.A.P., S.F.Y., C.B.N., J.W., J.A.M.A., N.L., and A.P.S. designed experiments. S.F.Y., C.B.N., J.W., M.F., R.L., J.A.M.A. and L.M. conducted experimentation and analysed data. V.J.M. and A.J.D. provided the purified RPA protein complex. J.P.M. provided the pentaprobe. A.J.C. provided the FUCCI cells. H.A.P. and S.F.Y. wrote the manuscript with editorial input from all authors.

## Competing interests

The authors declare no competing interests.
