## [Peer Review File · Nature Communications]

ZNF827 is a single-stranded DNA binding protein that regulates the ATR-CHK1 DNA damage response pathwayREVIEWER COMMENTS

Reviewer #1 (Remarks to the Author):

In this study, Yang and colleagues identify a previously unknown role for ZNF827 in the activation of the ATR-CHK1 DNA damage response. ZNF827 binds directly to ssDNA via two clusters of C2H2 zinc finger motifs and accumulates at sites of replicative damage, where it interacts with TOPBP1 to activate ATR, all in all leading to the engagement of HR-mediated repair. Additionally, the authors demonstrate that ZNF827 depletion disturbs replication initiation and replication fork progression, results in an accumulation of DNA damage, p21 induction, and a G1/S arrest. ZNF827 depletion also sensitized cancer cells to the chemotherapeutic agent topotecan, thereby highlighting the importance of ZNF827 as a novel druggable target within the ATR-CHK1 DNA damage response.

Overall, the data seem in line with the conclusions. The manuscript is very clearly written and has a logical flow. In addition, the function of ZNF827 in the genome-wide DDR is novel and important. As such, this manuscript will be of general interest for those interested in DNA replication and genome maintenance. However, important issues need to be addressed for sufficient support of this new role of ZNF827 and of the associated mechanism.

Major concerns:

1. Although the authors do EMSA's with purified proteins, they do copurify additional proteins. Hence it would be good if they support their EMSAs with including a supershift using a ZNF827 antibody, to convince that the gelshift/binding seen is caused by ZNF827, not by a copurifying protein (even though it is present in lower quantity).
2. Complementation experiments with WT and mutant ZNF827 are lacking and only 1 knockout line or siRNA for ZNF827 is used in the experiments. Main findings should be shown for multiple independent siRNAs and KO lines, along with complementation experiments to exclude off-target effects or effects being due to clonal variation, and to demonstrate an activity of ZNF827 instead of only assessing consequences of its loss. For instance the authors could assess whether the CHK1 activation defect in the ZNF827 knockout cells is rescued upon expression of the WT ZNF827, but not with the mutant ZNF827 that cannot bind ssDNA.
Also, multiple and independent ways of depleting ZNF827 are required to draw solid conclusions on the role of ZNF827 in HR (data are only for one siRNA). As also indicated above, the findings need to be supported by complementation of WT ZNF827 and ideally the mutant ZNF827 that cannot bind ssDNA, and through the use of multiple independent siRNAs.
3. The data supporting TOPBP1 recruitment to sites of replicative damage via ZNF827 is insufficient. A convincing experiment with a form of quantification, that shows that TOPBP1 is less recruited to sites of replicative damage in absence of ZNF827, is lacking. The authors could perform SIF analysis to detect the association of TOPBP1 to sites of replication stress in control and ZNF827-depleted cells, as well as in cells complemented with WT vs the ZNF1-3 mutant that does not bind ssDNA.
4. The authors use FANCM depletion as an alternative method to show the localization of ZNF827 to sites of replication stress. However, this involves assessing the localization of ZNF827 to telomeres. Ideally, the authors should perform a telomere-unrelated assay (e.g. localization to common fragile sites, SIF, IPOND), since this paper aims to show a new role for ZNF827 outside of telomeres.
5. The authors should revisit their statement that ZNF827 depletion has a stronger effect on ATR-CHK1 activation than the ATRi, given that total CHK1 levels are reduced upon ZNF827 depletion, while no reduction of total CHK1 levels is observed for the ATRi. In addition, see also below, there is no ZNF827 blot that shows equal depletion of ZNF827 across the conditions, which makes comparing effect sizes across conditions impossible/unreliable.

6. How can the authors' finding that TOPBP1 overexpression rescues/overcomes ZNF827 deficiency be reconciled with their proposed mechanism that TOPBP1 recruitment is via ZNF827? If ZNF827 is key to TOPBP1 recruitment, then in absence of ZNF827 there would be no recruitment of TOPBP1, regardless of whether TOPBP1 is overexpressed or not.

7. To further support the proposed model and understand the mechanism why TOPBP1 and not ETAA1 overexpression rescues ATR activation in ZNF827 depleted cells: Does ETAA1 (not) co-IP with ZNF827?

8. How specific is the ZNF827 antibody? Since many of the ZNF827 foci do not colocalize with ssDNA foci (fig. 1ca). Perhaps the authors could show a validation experiment of this antibody by performing a ZNF827 IF in the ZNF827 knockout cell line or ZNF827 depleted cells within the same experiment.

9. What is the effect of ZNF827 depletion on RPA foci formation? As this could also be highly relevant for the deficient ATR-checkpoint activation observed. In particular since e.g. in figure 1g RPA32 foci seem reduced/are less prominent in the ZNFdelta 1-3, ZNFdelta 4-9 and the truncated ZNF827 mutant forms. Quantifications would be important here (see also the more general point below regarding IF quantifications).

10. Association of ZNF827 to stalled forks is based by the authors on the recruitment of ZNF827 to sporadic discrete PCNA foci. This experiment does not very much convince. Authors should consider alternative assays to demonstrate localization of ZNF827 to stalled replication forks, e.g. by SIRF.

11. Line 307-308: 'Specifically, ZNF827 depletion resulted in decreased expression of CDC45 and MCM5 (Fig. 5J)'. However, the reduction in MCM5 levels is very mild or lacking (the only noticeable small drop minus topotecan seems to match the loading control vinculin, and no difference amongst samples with topotecan). The authors should provide quantification of these blots in 5J (from multiple independent experiments) to better support their statement on changes in the levels of any of the proteins that they think to change and where needed they should rephrase their statements.

12. How does ZNF827 work in HR? And how different if this HR role at the genome wide level in comparison to its role at telomeres? The authors could perhaps examine the Δ RRK mutant in the HR assays. In addition, the authors showed previously that NuRD-ZNF827 is able to recruit HR-related factors, such as the DDR protein BRIT1 and BRCA1. Are these/such factors also recruited by ZNF827 during HR-mediated repair of genome-wide replicative damage (outside telomeres)?

13. In multiple occasions (e.g. Figure 1c, 1g, 2a, 2f, S2a, 3e) quantifications of the IFs are lacking. Without the quantifications of a significant number of cells per replicate and over 3 or more independent biological experiments with statistics it is not possible to appreciate how robust and representative the data is.

14. Quantifications of key results shown by western blots are missing. Moreover, none of the blots (and other experiments) show the level of depletion/expression of ZNF827 protein. This is in particular essential to the rescue experiments in Figure 3C and 3D. Without knowing the ZNF827 protein levels across the different samples in these experiments it is impossible to conclude if TOPBP1 or ETAA1 overexpression rescue or not rescue the phenotype of ZNF827 depletion. Are ZNF827 protein levels equally down in the TOPBP1 overexpressing cells as in the controls?

15. The authors aim to claim that the new role of ZNF827 described here is independent of its NURD binding capacity, while in the telomeric role of ZNF827 described by the same group, recruitment of NURD by ZNF827 is important. To convince more on the genomic response to replication stress not involving NURD, it would help if Fig. 1D would have assessed also the ssDNA binding capacity of the

deltaRRK mutant, if experiments in Fig 1g would be appropriately quantified, and if authors would address whether NuRD is detected or not at sites of replicative damage in their experiments (in case it is, whether this is independent of ZNF827).

16. Do the authors have an explanation for why total CHK1 levels go down in ZNF827 depleted cells? Given that total CHK1 levels also go down, it makes it difficult to appreciate the extent of the CHK1 phosphorylation/activation defect. It would help very much if authors would quantify the phospho-CHK1 over total CHK1 over multiple independent experiments.

Minor concerns:

-Line 145: 'Addition of ZNF827 RPA to the ZNF827-ssDNA reaction', should probably be 'Addition of RPA to the ZNF827-ssDNA reaction'.

-Line 692: The formatting of this reference is not according to the rest of the references of the manuscript.

-Line 707: BrdU is first introduced in its abbreviated form, while in line 711 it is written as text and abbreviation.

Reviewer #2 (Remarks to the Author):

The authors report on the role of ZNF827 in the ATR-CHK1 DNA damage response pathway. The authors show that ZNF827 is an ssDNA binding protein, associated with the replication machinery that accumulates at stalled forks. Interestingly, ZNF827 interacts with TOPBP1, and cells depleted for ZNF827 are defective in ATR signaling and are sensitized to the topoisomerase inhibitor topotecan. Based on this data the authors propose a role for ZNF827 as a key modulator of the ATR-CHK1 DNA damage response pathway.

The data are clear, but several controls are required to fully substantiate the critical role of ZNF827 in the ATR-CHK1 DNA response pathway:

Major points:

- 1- The authors propose that ZNF827 recruit TOPBP1. However, no data in the support of this idea is provided. Furthermore, how do the authors reconcile this hypothesis with the fact that TOPBP1 overexpression can rescue some of the defects observed in cells depleted for ZNF827? Additional localization studies for ATR signaling molecules in the presence or absence of ZNF827 could provide additional mechanistic insights.
- 2- Upon ZNF827 depletion some cell lines induce CHK1 activation (HT1080) while others show a reduction in CHK1 activation (U2OS): how do the authors explain this difference?
- 3- Depletion of ZNF827 has a strong effect on cell proliferation (Fig 4), could these differences account for the impaired ATR activation/ differences in HR reporter assays?
- 4- The authors propose a central role for ZNF827 in an essential pathway, nevertheless ZNF827 depleted cells are viable (see in addition to this manuscript data from the Project Achilles). This is in contrast with what has been reported for proteins that play a critical role in the ATR/HR pathway. How do the authors explain the discrepancy between the proposed molecular function role and weak cellular phenotype?

Major point:

- 1- It is difficult sometimes to track which cells were used in any given experiment, consistent labeling within images would help the readers (see for example suppl figure 3 panel D).

Reviewer #3 (Remarks to the Author):

In this paper by Yang et al, the authors described a potentially novel function of the poorly characterized factor ZNF827 in replication stress response. They previously characterized a role of ZNF827 in homology dependent repair during the Alternative Lengthening of Telomeres (ALT). In this manuscript, they explored the genome-wide functions of ZNF827 and found ZNF827 binds to single-stranded DNA (ssDNA) through C2H2 zinc fingers motifs, associates with RPA in a ssDNA-dependent manner, and accumulates at a subset of DNA replication sites, where it interacts with TOPBP1 and ATR. They further showed that knockdown of ZNF827 compromised topoisomerase I inhibitor (topotecan) induced ATR activation and homology dependent repair. Moreover, depletion of ZNF827 impairs replication integrity, fork speed, causes cell cycle arrest at G1/S transition with induction of p21 protein. The DNA damage response pathway has been extensively studied in the past twenty years and leads to the identification of many new players and potential therapeutic targets. The characterization of ZNF827 as a new member of the DDR further expand on this knowledge and provides a potential new modulator of ATR kinase signaling pathway. However, whether this role of ZNF827 in replication stress response is truly independent from its role at telomeres, or it is secondary to telomere instability remain elusive and would be critical for establishing a novel function of ZNF827 in ATR mediated DNA damage response in general.

1) One important conclusion of the manuscript is ZNF827 traveling with replication fork, inferred mainly through colocalization with PCNA foci. This conclusion is supported by co-localization of ZNF827 with ssDNA (Fig 1c labeled with BrdU), RPA, and PCNA (Fig 2a). Yet, it seems that only a minor subset (potentially those on telomere, Figure 2F) of ZNF827 colocalized with ssDNA (Figure 1g) and ZNF827 only apparently co-localizes with PCNA in late S phase cells (not in early S phase). Given the known function of ZNF827 at telomeres, could the partial colocalization with BrdU (ssDNA), RPA and with PCNA reflects telomere binding, not replication fork? If ZNF827 is associated with replication fork in general, what does prevent ZNF827 from colocalizing with PCNA in early S phase cells? Colocalization experiments would also benefit from quantification, only representative images are reported for Fig 1c and 2a. The specificity of the ZNF827 foci in Figure 2d for ZNF827 and γ H2AX is also a concern, given significant green foci fall outside the DAPI staining in Figure 2d (but not in Figure 2a and Figure 1g)

Overall, additional evidence is needed to establish a telomere independent interaction of ZNF827 with replication forks. iPOND experiments, co-localization outside of telomere (with telomere probe), or interaction (direct or DNA-mediated) with bona fide core replication factors independent of DNA could be considered.

2) In Figure 2d the authors analyzed the effects of depleting ZNF827 on replication fork progression. They found new replication origins are affected by ZNF827 knockdown as well as moderate fork stalling. Since a general slowdown of the replication program could also reduce replication fork speed and origin firing, additional analyses are necessary to establish a "direct" role of ZNF827 on replication. Consistent with this concern, siZNF827 reduced the expression of S phase genes, including CHK1 (Figure 3c) and MCM5 (Figure 5J), which could indirect affect the response to Topotecan and the measurement for ATR signaling – CHK1p etc. How does the S phase (percentage, and BrdU incorporation rate) of these cells look like compared to control? Can ATR activation be measured in a controlled cell cycle environment? Whether ZNF827 affects replication fork recovery/restart – a function that has been attributed to ATR? This would help define a role of ZNF827 in ATR dependent events?

3) In Figure 3 the authors provide evidences that ZNF827 significantly affect the ATR/CHK1 pathway under topotecan treatment. It would be help to test whether this is specific formTOP1 inhibition. For instance, HU been extensively used to ctivate ATR pathway and CHK1 phosphorylation. These data

would pinpoint (or not) to a general requirement of ZNF827 in ATR activation following replication stress. Given the reduction of S phase specific proteins (CHK1/MCM/CDC45), it is critical to determine whether the lack of CHK1p can be explained by the reduced S phase cells.

In this context, authors conclude ZNF827 depletion and ATRi treatment have additive effect in further suppressing ATR signaling, referring to Figure 3b. But looking at CHK1-P and RPA S33-P in Figure 3b, the additive effect is not convincing. Can the authors provide additional or independent evidences/quantifications supporting this claim?

4) In supplementary Figure 3, the authors observed the effect of ATR-CHK1 pathway suppression upon ZNF827 depletion in several cell lines, with big variations in the phenotypes reported. Though this variation could be related to the different cell lines per se, could it be also due to different efficiency of ZNF827 depletion by siRNA? Western blot of ZNF827 protein levels would help to clarify the issues. Given the role of ZNF827 in ALT-telomere lengthening, it would also helpful to determine whether telomerase expression contribute to the phenotype variations.

5) In the last figures, authors detect several cell cycle defects associated with ZNF827 depletion and a striking overexpression of p21 inhibitor consistent with extended duration of G1 phase. When they went on analyzing the levels of some of the origin loading or replication factors following ZNF827 depletion, the phenotypes are not consistent. The decreased expression of CDC45 can be appreciated, but the claimed decreased expression of MCM5 is only reasonable without Topotecan and not clear after Topotecan (Western blot Figure 5j). And MCM6 or PCNA levels are not changed following ZNF827 knockdown, as acknowledged by the authors. Therefore, the sentence "we observed changes in expression of several components of the replication origin licensing and activation machinery following ZNF827 depletion" needs to be supported by additional experiments (maybe cell cycle distribution analyses and cell cycle specific western blot on S phase protein content) and quantifications (quantify independent western blots of MCM5 and CDC45 and other additional factors) or the conclusions need to be re-formulated based on the mild evidences provided.

RESPONSE TO REVIEWERS

Reviewer #1:

In this study, Yang and colleagues identify a previously unknown role for ZNF827 in the activation of the ATR-CHK1 DNA damage response. ZNF827 binds directly to ssDNA via two clusters of C2H2 zinc finger motifs and accumulates at sites of replicative damage, where it interacts with TOPBP1 to activate ATR, all in all leading to the engagement of HR-mediated repair. Additionally, the authors demonstrate that ZNF827 depletion disturbs replication initiation and replication fork progression, results in an accumulation of DNA damage, p21 induction, and a G1/S arrest. ZNF827 depletion also sensitized cancer cells to the chemotherapeutic agent topotecan, thereby highlighting the importance of ZNF827 as a novel druggable target within the ATR-CHK1 DNA damage response.

Overall, the data seem in line with the conclusions. The manuscript is very clearly written and has a logical flow. In addition, the function of ZNF827 in the genome-wide DDR is novel and important. As such, this manuscript will be of general interest for those interested in DNA replication and genome maintenance. However, important issues need to be addressed for sufficient support of this new role of ZNF827 and of the associated mechanism.

Major concerns:

1. Although the authors do EMSA's with purified proteins, they do copurify additional proteins. Hence it would be good if they support their EMSAs with including a supershift using a ZNF827 antibody, to convince that the gelshift/binding seen is caused by ZNF827, not by a copurifying protein (even though it is present in lower quantity).

Thank you for this suggestion. We have performed a super shift assay to demonstrate that the gel shift is attributable to ZNF827 binding. This experiment is now included as Figure S1D of the revised manuscript. ZNF827 binding to ssDNA is further supported by an immunoprecipitation experiment (now Figure S1E), in which a direct ZNF827 antibody is used to pull down ssDNA following incubation with purified protein, indicative of binding being specific to ZNF827.

2. Complementation experiments with WT and mutant ZNF827 are lacking and only 1 knockout line or siRNA for ZNF827 is used in the experiments. Main findings should be shown for multiple independent siRNAs and KO lines, along with complementation experiments to exclude off-target effects or effects being due to clonal variation, and to demonstrate an activity of ZNF827 instead of only assessing consequences of its loss. For instance the authors could assess whether the CHK1 activation defect in the ZNF827 knockout cells is rescued upon expression of the WT ZNF827, but not with the mutant ZNF827 that cannot bind ssDNA. Also, multiple and independent ways of depleting ZNF827 are required to draw solid conclusions on the role of ZNF827 in HR (data are only for one siRNA). As also indicated above, the findings need to be supported by complementation of WT ZNF827 and ideally the mutant ZNF827 that cannot bind ssDNA, and through the use of multiple independent siRNAs. Only one knockout clone was obtained following CRISPR-Cas9 knockout experiments in U-2 OS cells. This cell line was included in the SCE assay (Figure 4A and B) and demonstrates similar suppression of HR compared to parental cells following topotecan treatment as observed with siRNA-mediated depletion of ZNF827.

To address the need to evaluate the effects of multiple siRNAs targeting ZNF827, we have now included two additional siRNAs. The three siRNAs result in comparable levels of ZNF827

knockdown, and all suppress ATR-CHK1 kinase pathway activation in response to topotecan treatment. These data are now included in Figure S2B and Figure S3A of the revised manuscript.

Complementation experiments have also now been included to strengthen the role of ZNF827 in ATR-CHK1 kinase pathway activation. Specifically, we observed rescue of pCHK1 and pRPA levels in U-2 OS cells depleted for ZNF827, following overexpression of wild-type ZNF827. Importantly, rescue was not observed in cells overexpressing the ZNF827 Δ 1-3 DNA binding mutant. These data support the specificity of ZNF827 in ATR pathway activation. These experiments are included in the text and in Figure 3B of the revised manuscript.

3. The data supporting TOPBP1 recruitment to sites of replicative damage via ZNF827 is insufficient. A convincing experiment with a form of quantification, that shows that TOPBP1 is less recruited to sites of replicative damage in absence of ZNF827, is lacking. The authors could perform SIRF analysis to detect the association of TOPBP1 to sites of replication stress in control and ZNF827-depleted cells, as well as in cells complemented with WT vs the ZNF1-3 mutant that does not bind ssDNA.

We apologise for the confusion. Our data demonstrate that TOPBP1 co-immunoprecipitates with ZNF827, indicative of an association between the two proteins. However, our data do not directly demonstrate TOPBP1 is recruited to sites of replicative damage by ZNF827. The discussion was somewhat misleading, and has been modified to reflect that ZNF827 interacts with TOPBP1 at sites of replication stress, rather than directly recruiting TOPBP1.

4. The authors use FANCM depletion as an alternative method to show the localization of ZNF827 to sites of replication stress. However, this involves assessing the localization of ZNF827 to telomeres. Ideally, the authors should perform a telomere-unrelated assay (e.g. localization to common fragile sites, SIRF, IPOND), since this paper aims to show a new role for ZNF827 outside of telomeres.

To address this point, we treated cells with IR and used laser micro-irradiation to further demonstrate localisation of ZNF827 to sites of global DNA damage. IR resulted in a significant induction of ZNF827 and γ H2AX nuclear foci, and ZNF827 colocalised with both γ H2AX and RPA32 at laser stripes. These data are included as Figures 2E and S2G of the revised manuscript.

5. The authors should revisit their statement that ZNF827 depletion has a stronger effect on ATR-CHK1 activation than the ATRi, given that total CHK1 levels are reduced upon ZNF827 depletion, while no reduction of total CHK1 levels is observed for the ATRi. In addition, see also below, there is no ZNF827 blot that shows equal depletion of ZNF827 across the conditions, which makes comparing effect sizes across conditions impossible/unreliable.

We agree that the change in total CHK1 levels upon ZNF827 depletion adds complexity to the interpretation of the ATRi data. For this reason, these data and the accompanying text have been removed from the revised manuscript.

6. How can the authors' finding that TOPBP1 overexpression rescues/overcomes ZNF827 deficiency be reconciled with their proposed mechanism that TOPBP1 recruitment is via ZNF827? If ZNF827 is key to TOPBP1 recruitment, then in absence of ZNF827 there would be no recruitment of TOPBP1, regardless of whether TOPBP1 is overexpressed or not. To clarify, we do not believe that TOPBP1 is exclusively being recruited to sites of replication stress by ZNF827. Our data demonstrate that TOPBP1 co-immunoprecipitates with ZNF827, indicative of an association between the two proteins, but our data do not support direct

recruitment of TOPBP1 by ZNF827. Partial rescue of ATR signalling with TOPBP1 overexpression is consistent with ZNF827 and TOPBP1 being recruited to sites of replication stress independently of each other. This has been clarified in the text.

7. To further support the proposed model and understand the mechanism why TOPBP1 and not ETAA1 overexpression rescues ATR activation in ZNF827 depleted cells: Does ETAA1 (not) co-IP with ZNF827?

We have now performed a co-IP in both U-2 OS and HT1080 cells, demonstrating that ETAA1 does co-IP with ZNF827. These data are included in Figure 3G of the revised manuscript. We believe that ZNF827 is recruited to sites of replication stress independently of both TOPBP1 and ETAA1. It is not clear exactly why TOPBP1 overexpression partially rescues the ATR activation defect, while ETAA1 overexpression does not, but likely reflects the type of DNA damage and the different mechanisms by which these two proteins are recruited to DNA lesions.

8. How specific is the ZNF827 antibody? Since many of the ZNF827 foci do not colocalize with ssDNA foci (fig. 1ca). Perhaps the authors could show a validation experiment of this antibody by performing a ZNF827 IF in the ZNF827 knockout cell line or ZNF827 depleted cells within the same experiment.

Antibody validation using overexpressed myc-tagged ZNF827 is included in Figure S2C of the revised manuscript. We have now quantitated the data presented in Figure 1C. We believe that the discrepancies between ssDNA and ZNF827 staining may be attributable to a lack of sensitivity of BrdU-staining for ssDNA.

9. What is the effect of ZNF827 depletion on RPA foci formation? As this could also be highly relevant for the deficient ATR-checkpoint activation observed. In particular since e.g. in figure 1g RPA32 foci seem reduced/are less prominent in the ZNFdelta 1-3, ZNFdelta 4-9 and the truncated ZNF827 mutant forms. Quantifications would be important here (see also the more general point below regarding IF quantifications).

We have included quantitation of the colocalization data presented in Figure 1G. In addition, we have now performed IF experiments following ZNF827 depletion, and quantitated mean intensity of pRPA foci/RPA foci in S-phase cells, identified by EdU staining. We observed a significant decrease in pRPA foci/RPA foci following ZNF827 depletion in topotecan treated cells, consistent with defective ATR signalling caused by ZNF827 depletion. These new data are included in Figure 3C of the revised manuscript.

10. Association of ZNF827 to stalled forks is based by the authors on the recruitment of ZNF827 to sporadic discrete PCNA foci. This experiment does not very much convince. Authors should consider alternative assays to demonstrate localization of ZNF827 to stalled replication forks, e.g. by SIRF.

To address this point, we performed a co-immunoprecipitation experiment and demonstrated co-immunoprecipitation of ATRIP, the regulatory partner of ATR, and the 9-1-1 DNA damage sensor complex component Rad9, with ZNF827. These data have been included in Figure 3G and Figure S2A of the revised manuscript.

11. Line 307-308: 'Specifically, ZNF827 depletion resulted in decreased expression of CDC45 and MCM5 (Fig. 5J)'. However, the reduction in MCM5 levels is very mild or lacking (the only noticeable small drop minus topotecan seems to match the loading control vinculin, and no difference amongst samples with topotecan). The authors should provide quantification of these blots in 5J (from multiple independent experiments) to better support their statement on

changes in the levels of any of the proteins that they think to change and where needed they should rephrase their statements.

These results are reproducible, but their interpretation and how they fit with the overall story is not clear. For clarity, we have removed the blots of the replication origin licensing and activation machinery in response to ZNF827 depletion from the revised manuscript.

12. How does ZNF827 work in HR? And how different if this HR role at the genome wide level in comparison to its role at telomeres? The authors could perhaps examine the Δ RRK mutant in the HR assays. In addition, the authors showed previously that NuRD–ZNF827 is able to recruit HR-related factors, such as the DDR protein BRIT1 and BRCA1. Are these/such factors also recruited by ZNF827 during HR-mediated repair of genome-wide replicative damage (outside telomeres)?

We believe that ZNF827 plays a similar role at ALT telomeres compared to its role in genomic HR-mediated repair. This is consistent with the involvement of many well-defined genome-wide HR factors at ALT telomeres (for example BLM, MRN, RAD51, RAD52, etc). We have now performed complementation experiments that show rescue of the ATR activation defect with overexpression of wild-type ZNF827, but not with the ZNF827 Znf Δ 1-3 mutant. These data are now included as Figure 3B of the revised manuscript.

To determine whether ZNF827 functions to recruit HR factors to sites of replicative damage, we performed immunofluorescence to detect co-localisations between BRCA1 and pRPA in the context of ZNF827 depletion. Our data demonstrate that ZNF827 is required for BRCA1 localisation to sites of replication stress, consistent with ZNF827 recruiting HR factors to sites of replicative stress. These data are now included as Figure 4D-F of the revised manuscript.

Overall, our data demonstrate that ZNF827 nuclear localization, binding to ssDNA, and the interaction between ZNF827 and TOPBP1, are independent of ZNF827 binding to NuRD. This suggests that ZNF827 functions to bridge the genomic response to replication stress to the engagement of HR, potentially through NuRD-mediated recruitment of HR factors. This rationale has been strengthened in the discussion section of the revised manuscript.

13. In multiple occasions (e.g. Figure 1c, 1g, 2a, 2f, S2a, 3e) quantifications of the IFs are lacking. Without the quantifications of a significant number of cells per replicate and over 3 or more independent biological experiments with statistics it is not possible to appreciate how robust and representative the data is.

Quantitation and statistics for Figures 1C, 1G, and 2F have now been included. Figures 2A, S2A and 3E have since been removed.

14. Quantifications of key results shown by western blots are missing. Moreover, none of the blots (and other experiments) show the level of depletion/expression of ZNF827 protein. This is in particular essential to the rescue experiments in Figure 3C and 3D. Without knowing the ZNF827 protein levels across the different samples in these experiments it is impossible to conclude if TOPBP1 or ETAA1 overexpression rescue or not rescue the phenotype of ZNF827 depletion. Are ZNF827 protein levels equally down in the TOPBP1 overexpressing cells as in the controls?

We have now included quantitation of the key western blot data presented in Figure 3, and we have included ZNF827 antibody validation in Figure S2C of the revised manuscript. ZNF827 is a low abundance protein. The commercially available antibody is specific, but is unable to detect endogenous levels of ZNF827 via western blot. All western blots were performed on cells expressing exogenous ZNF827, and we have relied on qRT-PCR to quantitate knockdown

levels. We have now included qRT-PCR data in S2B of the revised manuscript to demonstrate robust ZNF827 knockdown with different siRNAs, in two cell lines, with and without topotecan treatment, and across different timepoints.

15. The authors aim to claim that the new role of ZNF827 described here is independent of its NURD binding capacity, while in the telomeric role of ZNF827 described by the same group, recruitment of NURD by ZNF827 is important. To convince more on the genomic response to replication stress not involving NURD, it would help if Fig. 1D would have assessed also the ssDNA binding capacity of the deltaRRK mutant, if experiments in Fig 1g would be appropriately quantified, and if authors would address whether NuRD is detected or not at sites of replicative damage in their experiments (in case it is, whether this is independent of ZNF827).

Quantitation of Figure 1G has now been included. These data demonstrate that ZNF827 nuclear localisation and ssDNA binding is independent of NuRD. Data presented in Figure 3F demonstrate that the interaction between ZNF827 and ATR/TOPBP1 is also independent of NuRD binding. We have now performed immunofluorescence experiments that show recruitment of BRCA1 to sites of replication stress is impeded upon ZNF827 depletion. These data implicate ZNF827 in bridging the genomic response to replication stress to the engagement of HR, potentially through NuRD-mediated recruitment of HR factors. These data are included in Figure 4D-F of the revised manuscript, and are discussed in the relevant sections of the results and discussion.

16. Do the authors have an explanation for why total CHK1 levels go down in ZNF827 depleted cells? Given that total CHK1 levels also go down, it makes it difficult to appreciate the extent of the CHK1 phosphorylation/activation defect. It would help very much if authors would quantify the phospho-CHK1 over total CHK1 over multiple independent experiments.

We believe this may be a cell cycle effect. We have now included quantitation of phospho-CHK1/total CHK1 throughout Figure 3. These data are consistent with our original interpretation and demonstrate a clear defect in CHK1 activation following ZNF827 depletion.

Minor concerns:

-Line 145: 'Addition of ZNF827 RPA to the ZNF827-ssDNA reaction', should probably be 'Addition of RPA to the ZNF827-ssDNA reaction'.

Corrected.

-Line 692: The formatting of this reference is not according to the rest of the references of the manuscript.

Corrected.

-Line 707: BrdU is first introduced in its abbreviated form, while in line 711 it is written as text and abbreviation.

Corrected.

Reviewer #2 (Remarks to the Author):

The authors report on the role of ZNF827 in the ATR-CHK1 DNA damage response pathway. The authors show that ZNF827 is an ssDNA binding protein, associated with the replication machinery that accumulates at stalled forks. Interestingly, ZNF827 interacts with TOPBP1, and cells depleted for ZNF827 are defective in ATR signaling and are sensitized to the

topoisomerase inhibitor topotecan. Based on this data the authors propose a role for ZNF827 as a key modulator of the ATR-CHK1 DNA damage response pathway.

The data are clear, but several controls are required to fully substantiate the critical role of ZNF827 in the ATR-CHK1 DNA response pathway:

Major points:

1- The authors propose that ZNF827 recruit TOPBP1. However, no data in the support of this idea is provided. Furthermore, how do the authors reconcile this hypothesis with the fact that TOPBP1 overexpression can rescue some of the defects observed in cells depleted for ZNF827? Additional localization studies for ATR signaling molecules in the presence or absence of ZNF827 could provide additional mechanistic insights.

We apologise for this confusion. We do not believe that ZNF827 directly recruits TOPBP1. Our data demonstrate that TOPBP1 co-immunoprecipitates with ZNF827, indicative of an association between the two proteins, but our data do not support direct recruitment of TOPBP1 by ZNF827. Furthermore, it is well established that TOPBP1 is recruited to ssDNA-dsDNA 5' junctions through interaction with MRN and the 9-1-1 checkpoint clamp. Independent recruitment of ZNF827 and TOPBP1 to sites of replicative stress is consistent with the partial rescue observed with TOPBP1. We have amended the text to clarify this point. In addition, we have performed further localization studies and demonstrated co-immunoprecipitation of both ATRIP, the regulatory partner of ATR, and the 9-1-1 complex component Rad9, with ZNF827. These data have been included in Figure 3G and Figure S2A of the revised manuscript.

2- Upon ZNF827 depletion some cell lines induce CHK1 activation (HT1080) while others show a reduction in CHK1 activation (U2OS): how do the authors explain this difference?

We believe our data consistently show a defect in CHK1 activation following ZNF827 depletion across multiple cell lines (U-2 OS, HT1080, HT1080 6TG, IICF/c). There is some variability in the timing following topotecan treatment at which we observe the reduced levels of CHK1, however the suppression is robust. We have now strengthened this result by including data for multiple ZNF827 siRNAs, showing consistent reduction in CHK1 activation following topotecan treatment. In addition, we have performed complementation experiments with wild-type and mutant ZNF827 overexpression, demonstrating specificity for the role of ZNF827 in ATR pathway activation. These data are presented in Figure S3A and Figure 3B of the revised manuscript.

3- Depletion of ZNF827 has a strong effect on cell proliferation (Fig 4), could these differences account for the impaired ATR activation/ differences in HR reporter assays?

We considered that cell cycle effects may contribute to the observed differences in HR. To address this, we analysed SCEs in U-2 OS CRISPR ZNF827 KO cells, which have no proliferative defect, and observed a striking defect in HR in the ZNF827 KO cells following treatment with topotecan. This defect was consistent with the HR defect observed in ZNF827 depleted cells. These data are included in Figure 4 and discussed in the relevant results section.

4- The authors propose a central role for ZNF827 is an essential pathway, nevertheless ZNF827 depleted cells are viable (see in addition to this manuscript data from the Project Achilles). This is in contrast with what has been reported for proteins that play a critical role in the ATR/HR pathway. How do the authors explain the discrepancy between the proposed molecular function role and weak cellular phenotype?

Our data demonstrate that ZNF827 plays an important role in ATR signalling and the HR pathway; however, we appreciate that ZNF827 is not an essential gene, consistent with other

proteins involved in the ATR/HR pathway, such as ETAA1 and 9-1-1 complex subunits that are also non-essential. To rationalise our findings, we believe that there may be subtleties in the DNA structure and/or specific DNA substrate that ZNF827 binds to that are beyond the scope of this project. This is something we are interested in and actively pursuing, but will take time to fully delineate.

Major point:

1- It is difficult sometimes to track which cells were used in any given experiment, consistent labeling within images would help the readers (see for example suppl figure 3 panel D). The cell lines used in Figure S3 have been labelled, and we have been mindful of labelling throughout.

Reviewer #3 (Remarks to the Author):

In this paper by Yang et al, the authors described a potentially novel function of the poorly characterized factor ZNF827 in replication stress response. They previously characterized a role of ZNF827 in homology dependent repair during the Alternative Lengthening of Telomeres (ALT). In this manuscript, they explored the genome-wide functions of ZNF827 and found ZNF827 binds to single-stranded DNA (ssDNA) through C2H2 zinc fingers motifs, associates with RPA in a ssDNA-dependent manner, and accumulates at a subset of DNA replication sites, where it interacts with TOPBP1 and ATR. They further showed that knockdown of ZNF827 compromised topoisomerase I inhibitor (topotecan) induced ATR activation and homology dependent repair. Moreover, depletion of ZNF827 impairs replication integrity, fork speed, causes cell cycle arrest at G1/S transition with induction of p21 protein. The DNA damage response pathway has been extensively studied in the past twenty years and leads to the identification of many new players and potential therapeutic targets. The characterization of ZNF827 as a new member of the DDR further expand on this knowledge and provides a potential new modulator of ATR kinase signaling pathway. However, whether this role of ZNF827 in replication stress response is truly independent from its role at telomeres, or it is secondary to telomere instability remain elusive and would be critical for establishing a novel function of ZNF827 in ATR mediated DNA damage response in general.

1) One important conclusion of the manuscript is ZNF827 traveling with replication fork, inferred mainly through colocalization with PCNA foci. This conclusion is supported by colocalization of ZNF827 with ssDNA (Fig 1c labeled with BrdU), RPA, and PCNA (Fig 2a). Yet, it seems that only a minor subset (potentially those on telomere, Figure 2F) of ZNF827 colocalized with ssDNA (Figure 1g) and ZNF827 only apparently co-localizes with PCNA in late S phase cells (not in early S phase). Given the known function of ZNF827 at telomeres, could the partial colocalization with BrdU (ssDNA), RPA and with PCNA reflects telomere binding, not replication fork? If ZNF827 is associated with replication fork in general, what does prevent ZNF827 from colocalizing with PCNA in early S phase cells? Colocalization experiments would also benefit from quantification, only representative images are reported for Fig 1c and 2a. The specificity of the ZNF827 foci in Figure 2d for ZNF827 and γ H2AX is also a concern, given significant green foci fall outside the DAPI staining in Figure 2d (but not in Figure 2a and Figure 1g)

We agree that ZNF827 and PCNA primarily colocalize in late S phase, indicative of ZNF827 accumulating at ssDNA present at stalled replication forks, rather than travelling with the replication fork. This point has been clarified in the revised manuscript. We have included quantitation of colocalization experiments throughout the manuscript, including Figure 1C. Figure 2A has now been removed. We have included ZNF827 antibody validation in Figure

S2C of the revised manuscript, and have replaced the images in Figure 2D with better quality representative images.

Overall, additional evidence is needed to establish a telomere independent interaction of ZNF827 with replication forks. iPOND experiments, co-localization outside of telomere (with telomere probe), or interaction (direct or DNA-mediated) with bona fide core replication factors independent of DNA could be considered.

We performed iPOND, but were unable to detect ZNF827 at nascently synthesised DNA. This is possibly due to low cellular levels of ZNF827 and the lack of sensitivity of the commercially available ZNF827 antibodies. We therefore used co-immunoprecipitation experiments to demonstrate interaction between ZNF827 and both ATRIP, the regulatory partner of ATR, and Rad9, which is a component of the 9-1-1 DNA damage sensor complex that play roles in the DNA damage response independent of telomeres. These data have been included in Figure 3G and Figure S2A of the revised manuscript.

2) In Figure 2d the authors analyzed the effects of depleting ZNF827 on replication fork progression. They found new replication origins are affected by ZNF827 knockdown as well as moderate fork stalling. Since a general slowdown of the replication program could also reduce replication fork speed and origin firing, additional analyses are necessary to establish a “direct” role of ZNF827 on replication. Consistent with this concern, siZNF827 reduced the expression of S phase genes, including CHK1 (Figure 3c) and MCM5 (Figure 5J), which could indirectly affect the response to Topotecan and the measurement for ATR signaling – CHK1p etc. How does the S phase (percentage, and BrdU incorporation rate) of these cells look like compared to control? Can ATR activation be measured in a controlled cell cycle environment? Whether ZNF827 affects replication fork recovery/restart – a function that has been attributed to ATR? This would help define a role of ZNF827 in ATR dependent events? This is an important consideration. To address this point, we have quantitated ATR activation by measuring the intensity of pRPA/RPA foci specifically in S-phase cells, which were detected by EdU incorporation. We observed no change in the number of pRPA foci per nucleus when comparing control versus siZNF827 cells, but we did observe a significant decrease in mean intensity of pRPA foci/RPA foci in the siZNF827 topotecan treated cells compared to control topotecan treated cells. This is indicative of the role of ZNF827 in ATR pathway activation being robust and independent of the cell cycle changes also conferred by ZNF827 depletion. These data have now been included in Figure 3C of the revised manuscript, and a paragraph describing these results has been added to the ATR-CHK1 kinase pathway activation section of the results.

3) In Figure 3 the authors provide evidences that ZNF827 significantly affect the ATR/CHK1 pathway under topotecan treatment. It would be help to test whether this is specific formTOP1 inhibition. For instance, HU been extensively used to ctivate ATR pathway and CHK1 phosphorylation. These data would pinpoint (or not) to a general requirement of ZNF827 in ATR activation following replication stress. Given the reduction of S phase specific proteins (CHK1/MCM/CDC45), it is critical to determine whether the lack of CHK1p can be explained by the reduced S phase cells.

Please see response to previous question. Quantitation of pRPA/RPA foci in EdU-labelled S-phase cells demonstrated a decrease in ATR activation in ZNF827 depleted cells following topotecan treatment (Figure 3C), indicative of the inhibition in ATR activation being independent of the reduced S phase cells. We have also treated cells with IR and used laser micro-irradiation to further demonstrate localisation of ZNF827 to sites of global DNA damage. IR resulted in a significant induction of ZNF827 and γ H2AX nuclear foci, and

ZNF827 colocalised with both γ H2AX and RPA32 at laser stripes. These data are included as Figures 2E and S2G of the revised manuscript.

In this context, authors conclude ZNF827 depletion and ATRi treatment have additive effect in further suppressing ATR signaling, referring to Figure 3b. But looking at CHK1-P and RPA S33-P in Figure 3b, the additive effect is not convincing. Can the authors provide additional or independent evidences/quantifications supporting this claim?

We agree that the effects of the ATRi treatments are difficult to interpret. These data and the accompanying text have been removed for clarity.

4) In supplementary Figure 3, the authors observed the effect of ATR-CHK1 pathway suppression upon ZNF827 depletion in several cell lines, with big variations in the phenotypes reported. Though this variation could be related to the different cell lines per se, could it be also due to different efficiency of ZNF827 depletion by siRNA? Western blot of ZNF827 protein levels would help to clarify the issues. Given the role of ZNF827 in ALT-telomere lengthening, it would also helpful to determine whether telomerase expression contribute to the phenotype variations.

The data in Figure S3 demonstrate suppression of CHK1 activation following ZNF827 depletion across multiple cell lines including both ALT (U-2 OS, IICF/c) and telomerase-positive (HT1080, HT1080 6TG) cell lines. There is some variability in the timing following topotecan treatment at which we observe the reduced levels of CHK1, however the suppression is robust. Unfortunately, the ZNF827 antibody is not sufficiently sensitive to detect endogenous levels of ZNF827. Consequently, we were unable to perform western blots to demonstrate ZNF827 knockdown and have relied on qRT-PCR for quantitation. Knockdown efficiency has now been included in Figure S2B of the revised manuscript.

5) In the last figures, authors detect several cell cycle defects associated with ZNF827 depletion and a striking overexpression of p21 inhibitor consistent with extended duration of G1 phase. When they went on analyzing the levels of some of the origin loading or replication factors following ZNF827 depletion, the phenotypes are not consistent. The decreased expression of CDC45 can be appreciated, but the claimed decreased expression of MCM5 is only reasonable without Topotecan and not clear after Topotecan (Western blot Figure 5j). And MCM6 or PCNA levels are not changed following ZNF827 knockdown, as acknowledged by the authors. Therefore, the sentence “we observed changes in expression of several components of the replication origin licensing and activation machinery following ZNF827 depletion” needs to be supported by additional experiments (maybe cell cycle distribution analyses and cell cycle specific western blot on S phase protein content) and quantifications (quantify independent western blots of MCM5 and CDC45 and other additional factors) or the conclusions need to be re-formulated based on the mild evidences provided.

A similar comment was raised by reviewer #1. We agree that these blots are not very convincing. The results are reproducible, but the changes in some of the replication-associated proteins are marginal and paradoxical, and the interpretation is not clear. For clarity, we have removed the blots of the replication origin licensing and activation machinery proteins in response to ZNF827 depletion.

REVIEWER COMMENTS

Reviewer #1 (Remarks to the Author):

This reviewer appreciates the efforts taken by the authors to respond to the comments made at the review of the first submission. The manuscript has significantly improved. A few issues remain with the revised manuscript.

-In response to comment 2, the authors aimed to strengthen their conclusions by including complementation experiments. However, figure 3B with such a complementation, remains difficult to interpret as there is no documentation of the knockdown levels of ZNF827 across the conditions, and there is no expression data shown of the exogenous complementation within that same experiment. Is there equal knockdown of endogenous ZNF827 across the control vector, WT and mutant ZNF827 reconstituted samples, and how does the protein expression level of the exogenous mutant ZNF827 compare to that of the exogenous WT ZNF827? If e.g. the mutant would be much lower expressed than the WT in this experiment, the lack of a rescue with the mutant is not informative, likewise if the knockdown is not the same across the conditions.

There is no indication of independent replication of this experiment (no n=?), making it unclear how robust the data is. No blots of repeated experiments or quantification of multiple experiments are shown, to deduce this from.

Also a comment regarding and/or explanation for how the authors interpret that the rescue with the WT is only partial is missing.

Minor, please note that the legend says that vinculin is used as loading control, while the blot is labeled with actin.

-Some concern still stays with the claims regarding the localization of ZNF827 at sites of replicative stress/stalled forks, unrelated to telomere localization (comments 4 and 10 of the original review). Many of the experiments are done in ALT positive cells, which complicates demonstrating a telomere unrelated localization, given the previous implication of ZNF827 in ALT. The new experiment 2E is again using an ALT+ cell line, with already a significant background of gH2AX foci. It is unclear to what extent the gH2AX/ZNF827 foci are colocalizing at telomeres or not. It is not clear why authors did not follow the suggestion to do a SIRF/PLA type of assay to try obtaining proof for physical association/local accumulation of ZNF827 to replication stress sites/stalled forks in ALT- cells (different from the previously described telomere association in ALT+ cells). Authors decided to do a co-IP for ZNF827 with ATRIP/Rad9 with benzonase treatment and upon topotecan treatment, again in ALT+ U2OS cells, and as already referred to above, they looked at colocalization of ZNF827 with gH2AX at IR or laser induced damage in another WI38-VA13 cell line (which if I am not mistaken is also ALT+). However, although indicative, such approaches are not directly addressing physical association of ZNF827 at stalled forks/replication stress sites (and unrelated to ALT telomeres), as detection of native DNA to mark sites of replication is not included.

-Several graphs/figures in the manuscript seem to miss clear indication of the number of independent replicate experiments that were done/included, making the robustness/reproducibility of the data uncertain. For instance, new figure 2E, seems to be quantification of only a single experiment where 50 nuclei were counted for colocalizing foci. A single experiment is not very informative. Biological replicates are needed to allow conclusions, ideally at least triplicates to allow meaningful statistical analysis.

-Several graphs still miss statistics: 2E, 4D-F.

-S2G doesn't have quantification.

Reviewer #2 (Remarks to the Author):

The authors addressed my concerns in full.

Reviewer #3 (Remarks to the Author):

The revised manuscript has addressed most of my concerns.

RESPONSE TO REVIEWERS

Reviewer #1:

This reviewer appreciates the efforts taken by the authors to respond to the comments made at the review of the first submission. The manuscript has significantly improved. A few issues remain with the revised manuscript.

1. In response to comment 2, the authors aimed to strengthen their conclusions by including complementation experiments. However, figure 3B with such a complementation, remains difficult to interpret as there is no documentation of the knockdown levels of ZNF827 across the conditions, and there is no expression data shown of the exogenous complementation within that same experiment. Is there equal knockdown of endogenous ZNF827 across the control vector, WT and mutant ZNF827 reconstituted samples, and how does the protein expression level of the exogenous mutant ZNF827 compare to that of the exogenous WT ZNF827? If e.g. the mutant would be much lower expressed than the WT in this experiment, the lack of a rescue with the mutant is not informative, likewise if the knockdown is not the same across the conditions.

Knockdown could not be measured directly in the complementation experiment, as there is no commercially available antibody to detect endogenous ZNF827 by western blot, meaning that we rely on qRT-PCR to confirm knockdown, which does not distinguish endogenous versus exogenous expression. In all other experiments, we saw robust knockdown of ZNF827. These data are included in Figure S2B. Protein expression of the exogenous wild-type and mutant ZNF827 are now included in Figures 3B, S3B and S3C.

There is no indication of independent replication of this experiment (no n=?), making it unclear how robust the data is. No blots of repeated experiments or quantification of multiple experiments are shown, to deduce this from.

The data are robust. The complementation experiment was repeated three times, with comparable results. The additional blots from the biological replicates are now included in Figures S3B and S3C.

Also a comment regarding and/or explanation for how the authors interpret that the rescue with the WT is only partial is missing.

A comment regarding the partial rescue with wild-type ZNF827 has now been included.

Minor, please note that the legend says that vinculin is used as loading control, while the blot is labeled with actin.

The figure legend has been changed to reflect the use of actin as the loading control.

2. Some concern still stays with the claims regarding the localization of ZNF827 at sites of replicative stress/stalled forks, unrelated to telomere localization (comments 4 and 10 of the original review). Many of the experiments are done in ALT positive cells, which complicates demonstrating a telomere unrelated localization, given the previous implication of ZNF827 in ALT. The new experiment 2E is again using an ALT+ cell line, with already a significant background of gH2AX foci. It is unclear to what extent the gH2AX/ZNF827 foci are colocalizing at telomeres or not. It is not clear why authors did not follow the suggestion to do a SIRF/PLA type of assay to try obtaining proof for physical association/local accumulation of ZNF827 to replication stress sites/stalled forks in ALT- cells (different from the previously described telomere association in ALT+ cells). Authors decided to do a co-IP for ZNF827 with

ATRIP/Rad9 with benzonase treatment and upon topotecan treatment, again in ALT+ U2OS cells, and as already referred to above, they looked at colocalization of ZNF827 with γ H2AX at IR or laser induced damage in another WI38-VA13 cell line (which if I am not mistaken is also ALT+). However, although indicative, such approaches are not directly addressing physical association of ZNF827 at stalled forks/replication stress sites (and unrelated to ALT telomeres), as detection of native DNA to mark sites of replication is not included.

We did not attempt PLA, because unfortunately we did not have the right antibody combination. Nevertheless, we have shown using numerous approaches that ZNF827 localises to sites of replication stress. This includes association of ZNF827 with RPA (Figures 1E-G and S1G; Supp Info 1), co-IP of ZNF827 with ATR, ATRIP and Rad9 (Figure S2A; Figure 3G), localisation of ZNF827 to sites of replication stress following topotecan treatment (Figure 2C; Figure S2D-E), and localisation of ZNF827 with RPA and γ H2AX at laser stripes (Figure S2G).

Furthermore, the majority of the experiments in the manuscript that show localisation of ZNF827 to sites of replication stress (Figures S1G, S2D-E, S2G), the role of ZNF827 in the replication stress response (Figures S3A, S3D-E), how ZNF827 contributes to HR (Figures 4B, 4D-F, 4H), and cell cycle dysregulation (Figures 5D-I), were performed in ALT-negative (HT1080 or HT1080 6TG) cells, comprehensively demonstrating ZNF827 function beyond its role at ALT telomeres. This includes the laser micro-irradiation experiment in Figure S2G, which was performed in HT1080 ALT-negative cells. For clarity, we have removed Figure 2E.

3. Several graphs/figures in the manuscript seem to miss clear indication of the number of independent replicate experiments that were done/included, making the robustness/reproducibility of the data uncertain. For instance, new figure 2E, seems to be quantification of only a single experiment where 50 nuclei were counted for colocalizing foci. A single experiment is not very informative. Biological replicates are needed to allow conclusions, ideally at least triplicates to allow meaningful statistical analysis.

Figure 2E has been removed.

4. Several graphs still miss statistics: 2E, 4D-F.

Figure 2E has been removed. Statistics have been included for Figures 4D-F.

5. S2G doesn't have quantification.

Quantitation of Figure S2G has been included.

Reviewer #2:

The authors addressed my concerns in full.

Reviewer #3:

The revised manuscript has addressed most of my concerns.